# BIG LEARNING VARIATIONAL AUTO-ENCODERS

## ABSTRACT

As a representative latent variable model, the Variational Auto-Encoder (VAE) is powerful in modeling high-dimensional signals like images and texts. However, practical applications often require versatile data capabilities, such as conditional generation/completion, inference with incomplete/marginal data, *etc*, which are challenging to harvest from a conventional/joint VAE. To satisfy those requirements, we leverage the recently proposed big learning to upgrade the joint VAE to its big-learning variant termed BigLearn-VAE, which delivers joint, marginal, and conditional generation/completion, inference, and reconstruction capabilities, simultaneously. In addition, we also reveal that the BigLearn-VAE can be constructed based on one foundation model, manifested as one universal model possessing plenty of versatile capabilities. Code will be released.

## 1 INTRODUCTION

The variational auto-encoder (VAE) Kingma & Welling (2014); Rezende et al. (2014) is a representative latent variable generative model that elegantly combines powerful deep neural networks (NN) with principled variational inference. Thanks to its stable training and versatile modeling capability brought by the latent code, the VAE and its many variants have found applications in a broad range of contexts, such as unsupervised/semi-supervised learning Izmailov et al. (2020), pseudo replay and data augmentation van de Ven & Tolias (2018); Norouzi et al. (2020), anomaly detection Park et al. (2022), text, image, and video generation Bowman et al. (2015); Razavi et al. (2019); Yan et al. (2021), audio and music synthesis Dhariwal et al. (2020); Kim et al. (2021), molecular processes Lim et al. (2018); Gómez-Bombarelli et al. (2018), and healthcare Han et al. (2019).

Despite being widely utilized, the conventional jointly-trained VAEs often fail to satisfy diverse conditional-sampling requirements, which are of high value and frequently arise from practical applications that are associated with missing data imputation, conditional data generation/completion, in-paining, recommendation, inference/feature-extraction with incomplete data, *etc*. On the other hand, to harvest those conditional sampling capabilities from a joint VAE via post-processing is generally intractable Simkus & Gutmann (2023) and existing techniques resort to the computationally expensive Gibbs/MCMC sampling Rezende et al. (2014); Mattei & Frellsen (2018); Simkus & Gutmann (2023).

To endow VAEs with versatile conditional sampling capabilities, we take inspiration from the recent ground-breakingly successful foundation models Bommasani et al. (2021); Yuan et al. (2022); Ouyang et al. (2022); Ramesh et al. (2022); Saharia et al. (2022); Rombach et al. (2022), or, more specifically, the big learning principle Cong & Zhao (2022) that contains most training objectives of foundation models as special cases. The big learning proposes to exhaustively exploit the information inherent in its large-scale training data, by simultaneously modeling many/all joint, conditional, and marginal data distributions across potentially diverse domains Cong & Zhao (2022).

The key insight is that the data samples demonstrating versatile conditional sampling capabilities (as well as joint and marginal sampling ones) are already present in the training data; accordingly, one should straightforwardly leverage those samples to form the corresponding capabilities. Note ideally, *perfect* joint training (*i.e.,* learning only with the joint data sample) is expected to implicitly deliver *perfect* conditional sampling capabilities, despite their extraction may be computationally expensive. More importantly, that *perfect* joint training is likely intractable in practice. Throughout the paper, all capabilities that can be demonstrated via manipulating the data constitute the *data capabilities*.

To explicitly endow VAEs with versatile data capabilities, we propose to leverage the big learning principle to upgrade the conventional jointly-trained VAE to its big-learning variant termed the big-learning VAE (BigLearn-VAE), which delivers joint, marginal, and conditional generation, inference, and reconstruction capabilities, simultaneously. In addition, based on general analyses on VAE modeling, we also reveal that the BigLearn-VAE can be constructed based on one universal foundation model, manifested as one model possessing plenty of versatile data capabilities.

Our contributions are summarized as follows.

- We propose the BigLearn-VAE, which upgrades the conventional jointly-trained VAE with comparable/better performance and versatile data capabilities like conditional sampling.
- We present general analyses on VAE modeling, which motivate us to develop the BigLearn-VAE based on one universal model; we empirically justify its effectiveness.
- We empirically demonstrate the big-learned data capabilities, particularly those associated with incomplete data, including inference/feature-extraction with incomplete data and arbitrary data-completion/in-painting.

## 2 PRELIMINARY

Below we briefly review the preliminary Variational Auto-Encoder (VAE) Kingma & Welling (2014); Rezende et al. (2014) and the big learning principle Cong & Zhao (2022) that lays the foundation of the presented techniques.

### 2.1 VARIATIONAL AUTO-ENCODERS

Unifying the power of deep neural networks (NN) and principled variational inference for latent variable generative models, the Variational Auto-Encoder (VAE) Kingma & Welling (2014); Rezende et al. (2014) learns a NN-parameterized generative model $p_{\boldsymbol{\theta}}(\boldsymbol{x}, \boldsymbol{z}) = p_{\boldsymbol{\theta}}(\boldsymbol{x}|\boldsymbol{z})p_{\boldsymbol{\theta}}(\boldsymbol{z})$, with a latent variable $\boldsymbol{z}$, to model the generative process of the complete/joint data sample $\boldsymbol{x}$ from the underlying data distribution $q(\boldsymbol{x})$, by maximizing the joint evidence lower bound (JointELBO) Jordan et al. (1999) of the intractable log-likelihood $\log p_{\boldsymbol{\theta}}(\boldsymbol{x})$, *i.e.,*

$$\text{JointELBO}(\boldsymbol{\theta}, \boldsymbol{\phi}) = \mathbb{E}_{q_{\phi}(\boldsymbol{z}|\boldsymbol{x})} \log p_{\boldsymbol{\theta}}(\boldsymbol{x}|\boldsymbol{z}) - \beta \text{KL}[q_{\phi}(\boldsymbol{z}|\boldsymbol{x})||p_{\boldsymbol{\theta}}(\boldsymbol{z})] \leq \log p_{\boldsymbol{\theta}}(\boldsymbol{x}), \qquad (1)$$

where $\beta = 1$, $q_{\phi}(\boldsymbol{z}|\boldsymbol{x})$ is a NN-parameterized variational inference arm that approximates the posterior $p_{\boldsymbol{\theta}}(\boldsymbol{z}|\boldsymbol{x})$, and KL represents the Kullback-Leibler (KL) divergence. Considering various practical applications Castrejon et al. (2019); Bae et al. (2022), $\beta$ is frequently treated as an important tunable hyperparameter, leading to the $\beta$-VAE Higgins et al. (2016); we also assume $\beta$ being a hyperparameter by default. Often the decoding $p_{\boldsymbol{\theta}}(\boldsymbol{x}|\boldsymbol{z}) = \mathcal{N}(\boldsymbol{x}|\boldsymbol{\mu}_{\boldsymbol{\theta}}(\boldsymbol{z}), \mathbf{I})$ is modeled with a "decoder" NN $\boldsymbol{\mu}_{\boldsymbol{\theta}}(\boldsymbol{z})$, the prior is specified as $p_{\boldsymbol{\theta}}(\boldsymbol{z}) = \mathcal{N}(\boldsymbol{z}|\mathbf{0}, \mathbf{I})$, and the encoding inference arm $q_{\phi}(\boldsymbol{z}|\boldsymbol{x}) = \mathcal{N}(\boldsymbol{z}|\boldsymbol{\mu}_{\phi}(\boldsymbol{x}), \text{diag}(\boldsymbol{\sigma}_{\phi}^2(\boldsymbol{x})))$ is constructed with an "encoder" NN that outputs both mean $\boldsymbol{\mu}_{\phi}(\boldsymbol{x})$ and standard deviation $\boldsymbol{\sigma}_{\phi}(\boldsymbol{x})$.

### 2.2 BIG LEARNING

Foundation models have brought many ground-breaking successes to diverse research fields Stickland & Murray (2019); Brown et al. (2020); He et al. (2021); Bommasani et al. (2021); Yuan et al. (2022); Ramesh et al. (2022); OpenAI (2022); Ouyang et al. (2022); OpenAI (2023); Touvron et al. (2023); Chowdhery et al. (2022), benefiting from both the valuable information within their large-scale (pre-)training data and the exhaustive exploitation of that information via comprehensive diverse (pre-)training. Recently, Cong & Zhao (2022) summarizes most of the pretraining strategies of foundation models from the generative perspective and condenses them into a unified big learning principle, as defined below.

**Definition 1** ((Uni-modal) big learning Cong & Zhao (2022)). *Given data samples* $\boldsymbol{x} \in \mathbb{R}^{L \times D}$ *from the underlying data distribution* $q(\boldsymbol{x})$, *with length L, dimension D, the length index set* $\mathbb{L} = \{1, \cdots, L\}$, *and any two non-overlapping subsets* $\mathbb{S} \subset \mathbb{L}, \mathbb{T} \subseteq \mathbb{L}, \mathbb{T} \neq \emptyset$, *the (uni-modal) big learning leverages a universal foundation model* $p_{\boldsymbol{\theta}}(\boldsymbol{x}_{\mathbb{T}}|\boldsymbol{x}_{\mathbb{S}}), \forall(\mathbb{S}, \mathbb{T})$ *to model many/all joint, conditional, and marginal data distributions simultaneously,* i.e.,

$$p_{\boldsymbol{\theta}}(\boldsymbol{x}_{\mathbb{T}}|\boldsymbol{x}_{\mathbb{S}}) \longrightarrow q(\boldsymbol{x}_{\mathbb{T}}|\boldsymbol{x}_{\mathbb{S}}), \forall(\mathbb{S}, \mathbb{T}) \in \boldsymbol{\Omega}, \qquad (2)$$

*where the arrow indicates utilizing its left-hand side to model its right-hand side. The actual objective measuring the distance/divergence (or encouraging the matching) between both sides of the arrow should be selected base on the application. $\boldsymbol{\Omega}$ is a user-defined set that contains the $(\mathbb{S}, \mathbb{T})$ pairs of interest. With different $(\mathbb{S}, \mathbb{T})$ pairs, $q(\boldsymbol{x}_{\mathbb{T}}|\boldsymbol{x}_{\mathbb{S}})$ may represent a joint/marginal/conditional data distribution, whose samples are readily available from the training data.*

The sample $\boldsymbol{x} \in \mathbb{R}^{L \times D}$ may represent $(i)$ a sentence with $L$ words and a vocabulary of size $D$; each row of $\boldsymbol{x}$ is then a $D$-dimensional one-hot vector, and $(ii)$ an image that is patchified as $L$ small patches of dimension $D$.

The core idea of the big learning is to keep consistency with the ideal situation. By ignoring any constraint temporarily, ideally there is an underlying analytical expression for joint $q(\boldsymbol{x})$, from which expressions for all conditional $q(\boldsymbol{x}_{\mathbb{T}}|\boldsymbol{x}_{\mathbb{S}})$ and marginal $q(\boldsymbol{x}_{\mathbb{T}})$ can be derived. Next, assume one has obtained a powerful model $p_{\boldsymbol{\theta}^*}(\boldsymbol{x})$ that *perfectly* matches $q(\boldsymbol{x})$; then the derived conditional $p_{\boldsymbol{\theta}^*}(\boldsymbol{x}_{\mathbb{T}}|\boldsymbol{x}_{\mathbb{S}})$ and marginal $p_{\boldsymbol{\theta}^*}(\boldsymbol{x}_{\mathbb{T}})$, with the same set of parameters $\boldsymbol{\theta}^*$, *perfectly* matches $q(\boldsymbol{x}_{\mathbb{T}}|\boldsymbol{x}_{\mathbb{S}})$ and $q(\boldsymbol{x}_{\mathbb{T}})$, respectively. The big learning explicitly employs one universal set of parameters (or a foundation model) and explicitly encourages those matchings simultaneously.

## 3 BIG LEARNING VARIATIONAL AUTO-ENCODERS

Below we follow the big learning principle in Definition 1 to upgrade the joint VAE with (1) into the Big Learning VAE (BigLearn-VAE). We begin with a general analysis on the modeling of VAEs, based on which we then present the upgraded BigLearn-VAE.

### 3.1 A GENERAL ANALYSIS ON THE MODELING OF VARIATIONAL AUTO-ENCODERS

Given a collection of data samples $\boldsymbol{x}$ from the underlying data distribution $q(\boldsymbol{x})$, the ultimate goal of the VAE is to learn a parameterized model $p_{\boldsymbol{\theta}^*}(\boldsymbol{x}) = \int p_{\boldsymbol{\theta}^*}(\boldsymbol{x}, \boldsymbol{z}) d\boldsymbol{z}$ to *perfectly* match $q(\boldsymbol{x})$, via maximum log-likelihood learning, *i.e.,*

$$\boldsymbol{\theta}^* = \underset{\boldsymbol{\theta}}{\operatorname{argmax}}\, \mathbb{E}_{q(\boldsymbol{x})} \log p_{\boldsymbol{\theta}}(\boldsymbol{x}) = \underset{\boldsymbol{\theta}}{\operatorname{argmin}}\, \mathrm{KL}[q(\boldsymbol{x})||p_{\boldsymbol{\theta}}(\boldsymbol{x})], \tag{3}$$

where the objective is equivalent to (2) with $\mathbb{S} = \emptyset$, $\mathbb{T} = \mathbb{L}$, and the arrow employing the KL divergence.

Next, we elaborate on a detailed analysis within both $\boldsymbol{x}$-space and $(\boldsymbol{x}, \boldsymbol{z})$-space, respectively.

- **$\boldsymbol{x}$-space** Assume infinite data samples, infinite model capacity of $p_{\boldsymbol{\theta}}(\boldsymbol{x})$, and infinite searching capability of the optimizer; then, ideally, one can get the optimal $p_{\boldsymbol{\theta}^*}(\boldsymbol{x}) = q(\boldsymbol{x}), \forall \boldsymbol{x}$. As mentioned earlier, a perfect joint model $p_{\boldsymbol{\theta}^*}(\boldsymbol{x})$ implicitly delivers perfect conditional and marginal matchings, *i.e.,* $p_{\boldsymbol{\theta}^*}(\boldsymbol{x}_{\mathbb{T}}|\boldsymbol{x}_{\mathbb{S}}) = q(\boldsymbol{x}_{\mathbb{T}}|\boldsymbol{x}_{\mathbb{S}})$ and $p_{\boldsymbol{\theta}^*}(\boldsymbol{x}_{\mathbb{T}}) = q(\boldsymbol{x}_{\mathbb{T}})$, respectively. Even though the analytical calculations of $p_{\boldsymbol{\theta}^*}(\boldsymbol{x}) = \int p_{\boldsymbol{\theta}^*}(\boldsymbol{x}, \boldsymbol{z}) d\boldsymbol{z}$, $p_{\boldsymbol{\theta}^*}(\boldsymbol{x}_{\mathbb{T}}) = \int p_{\boldsymbol{\theta}^*}(\boldsymbol{x}) d\boldsymbol{x}_{\mathbb{T}^{\complement}}$, and $p_{\boldsymbol{\theta}^*}(\boldsymbol{x}_{\mathbb{T}}|\boldsymbol{x}_{\mathbb{S}}) = p_{\boldsymbol{\theta}^*}(\boldsymbol{x}_{\mathbb{T} \cup \mathbb{S}})/p_{\boldsymbol{\theta}^*}(\boldsymbol{x}_{\mathbb{S}})$ are generally intractable, they undoubtedly share the same set of parameters $\boldsymbol{\theta}^*$ with the aforementioned ideal assumptions.

  However, by considering practical constraints of finite data sample, model capacity, and searching capability, it's likely that one gets an sub-optimal $p_{\hat{\boldsymbol{\theta}}}(\boldsymbol{x}) \neq q(\boldsymbol{x})$; accordingly, the post-calculated $p_{\hat{\boldsymbol{\theta}}}(\boldsymbol{x}_{\mathbb{T}})$, $p_{\hat{\boldsymbol{\theta}}}(\boldsymbol{x}_{\mathbb{T}}|\boldsymbol{x}_{\mathbb{S}})$, and their sampling capabilities are not reliable. Under such practical situations, one would prefer simultaneous joint, conditional, and marginal matchings with a shared universal model, *i.e.,* the big learning in Definition 1, which is expected to directly deliver *trained* data capabilities like conditional sampling and, at the same time, to encourage searching for a more reliable $\theta$ that is closer to $\theta^*$ in the sense of diverse joint/conditional/marginal matchings.

- **$(\boldsymbol{x}, \boldsymbol{z})$-space** Two different high-dimensional distributions, *i.e.,* $p_1(\boldsymbol{x}, \boldsymbol{z}_1)$ and $p_2(\boldsymbol{x}, \boldsymbol{z}_2)$, can share the same low-dimensional distribution $p(\boldsymbol{x})$. That means the aforementioned optimal $p_{\boldsymbol{\theta}^*}(\boldsymbol{x}) = q(\boldsymbol{x})$ can be satisfied by many different high-dimensional models $p_{\boldsymbol{\theta}_i^*}(\boldsymbol{x}, \boldsymbol{z}_i), i = 1, 2, \cdots$. Therefore, it's impossible to learn a *disentangled* $\boldsymbol{z}$-space in an unsupervised manner with only $\boldsymbol{x}$-information Locatello et al. (2019). Introducing a suitable prior for the latent $\boldsymbol{z}$-space is important and many works have been proposed Tomczak & Welling (2018); Casale et al. (2018); Davidson et al. (2018); Takahashi et al. (2019); Joo et al. (2020). How to specify a suitable $\boldsymbol{z}$-prior is left as future research and here we focus on leveraging the big learning principle to upgrade the VAE in the $\boldsymbol{x}$-space.

Based on the above analysis, we next elaborate on how to leverage the big learning principle in Definition 1 to develop the upgraded BigLearn-VAE.

## 3.2 On Introducing Big Learning in the $x$-Space

When given a complete/joint data sample $\boldsymbol{x} \sim q(\boldsymbol{x})$, one simultaneously receives a conditional sample for each conditional distribution $q(\boldsymbol{x}_{\mathbb{T}}|\boldsymbol{x}_{\mathbb{S}}), \forall(\mathbb{S}, \mathbb{T})$ and a marginal data sample for each marginal distribution $q(\boldsymbol{x}_{\mathbb{T}}), \forall \mathbb{T}$. However, a conventional/joint VAE only utilizes the joint data sample via the joint learning in (1), resulting in the under-utilization of those numerous diverse conditional and marginal data samples. Generally, we can not count on joint learning to automatically complete conditional/marginal matchings for us, especially when the model capacity is limited.

Different from the conventional jointly-trained VAE, we propose to leverage the big learning principle Cong & Zhao (2022) to explicitly and exhaustively exploit the information in joint, conditional, and marginal data samples, via simultaneous joint, conditional, and marginal matchings.

Below we first elaborate on the diverse matching tasks that constitute the big learning objective. Then, we present the modified model architectures that are compliant to that objective. Finally, combining the big learning objective and modified architectures, we deliver the proposed BigLearn-VAE.

### 3.2.1 Matching Tasks That Constitute the Big Learning Objective

The big learning objective consists of diverse joint, conditional, and marginal matching tasks. Specifically,

- **Joint matching of $p_{\boldsymbol{\theta}}(\boldsymbol{x}) \longrightarrow q(\boldsymbol{x})$.** Same with the conventional VAE Kingma & Welling (2014); Rezende et al. (2014); Higgins et al. (2016), the joint matching is conducted by maximizing the JointELBO in (1).

- **Marginal matching of $p_{\boldsymbol{\theta}}(\boldsymbol{x}_{\mathbb{T}}) \longrightarrow q(\boldsymbol{x}_{\mathbb{T}})$.** Because of the integral *w.r.t.* $\boldsymbol{z}$ and $\boldsymbol{x}_{\mathbb{T}^{\complement}}$, it's intractable to calculate the marginal $p_{\boldsymbol{\theta}}(\boldsymbol{x}_{\mathbb{T}}) = \iint p_{\boldsymbol{\theta}}(\boldsymbol{x}, \boldsymbol{z}) d\boldsymbol{z} d\boldsymbol{x}_{\mathbb{T}^{\complement}}$. Fortunately, the marginal generative process is often readily available from the modeling of $p_{\boldsymbol{\theta}}(\boldsymbol{x}, \boldsymbol{z})$.

  For example, given the parameterized joint generative process $p_{\boldsymbol{\theta}}(\boldsymbol{x}, \boldsymbol{z}) = p_{\boldsymbol{\theta}}(\boldsymbol{x}|\boldsymbol{z})p_{\boldsymbol{\theta}}(\boldsymbol{z})$ with conditionally independent $p_{\boldsymbol{\theta}}(\boldsymbol{x}|\boldsymbol{z}) = \mathcal{N}(\boldsymbol{x}|\boldsymbol{\mu}_{\boldsymbol{\theta}}(\boldsymbol{z}), \mathbf{I})$, the corresponding marginal generative process is analytically expressed as $p_{\boldsymbol{\theta}}(\boldsymbol{x}_{\mathbb{T}}, \boldsymbol{z}) = p_{\boldsymbol{\theta}}(\boldsymbol{x}_{\mathbb{T}}|\boldsymbol{z})p_{\boldsymbol{\theta}}(\boldsymbol{z})$, where $p_{\boldsymbol{\theta}}(\boldsymbol{x}_{\mathbb{T}}|\boldsymbol{z}) = \mathcal{N}(\boldsymbol{x}_{\mathbb{T}}|\boldsymbol{\mu}_{\boldsymbol{\theta}}(\boldsymbol{z})_{\mathbb{T}}, \mathbf{I}_{\mathbb{T}\mathbb{T}})$ and $\mathbf{I}_{\mathbb{T}\mathbb{S}}$ is the sub-matrix consisting of the $\mathbb{T}$ rows and $\mathbb{S}$ columns of $\mathbf{I}$. With the analytical $p_{\boldsymbol{\theta}}(\boldsymbol{x}_{\mathbb{T}}, \boldsymbol{z})$ that is derived from the parameterized $p_{\boldsymbol{\theta}}(\boldsymbol{x}, \boldsymbol{z})$, the marginal matching of $p_{\boldsymbol{\theta}}(\boldsymbol{x}_{\mathbb{T}}) \longrightarrow q(\boldsymbol{x}_{\mathbb{T}})$ can be similarly performed by maximizing the marginal ELBO (MarginELBO), which is defined as

$$\text{MarginELBO}(\boldsymbol{\theta}, \mathcal{M}) = \mathbb{E}_{q(\boldsymbol{x}_{\mathbb{T}})}\big[\mathbb{E}_{q_{\mathcal{M}}(\boldsymbol{z}|\boldsymbol{x}_{\mathbb{T}})} \log p_{\boldsymbol{\theta}}(\boldsymbol{x}_{\mathbb{T}}|\boldsymbol{z}) - \beta \text{KL}[q_{\mathcal{M}}(\boldsymbol{z}|\boldsymbol{x}_{\mathbb{T}})||p_{\boldsymbol{\theta}}(\boldsymbol{z})]\big], \quad (4)$$

  where $q_{\mathcal{M}}(\boldsymbol{z}|\boldsymbol{x}_{\mathbb{T}})$ is the marginal inference arm with its optimal being $q_{\mathcal{M}}^*(\boldsymbol{z}|\boldsymbol{x}_{\mathbb{T}}) = p_{\boldsymbol{\theta}}(\boldsymbol{z}|\boldsymbol{x}_{\mathbb{T}})$. Note intuitively, for different $\mathbb{T}$, one should define different parameterized inference arms, which is cumbersome. We will leverage our follow-up analysis to enable utilizing a universal foundation model to simultaneously model all marginal inference arms.

- **Conditional matching of $p_{\boldsymbol{\theta}}(\boldsymbol{x}_{\mathbb{T}}|\boldsymbol{x}_{\mathbb{S}}) \longrightarrow q(\boldsymbol{x}_{\mathbb{T}}|\boldsymbol{x}_{\mathbb{S}})$.** Similar to joint and marginal matchings, we again resort to the conditional variant to the vanilla ELBO, *i.e.,* the conditional ELBO (ConditionELBO), to perform the conditional matching task. Specifically,

$$\text{ConditionELBO}(\boldsymbol{\theta}, \mathcal{C}) = \mathbb{E}_{q(\boldsymbol{x}_{\mathbb{S} \cup \mathbb{T}})} \begin{bmatrix} \mathbb{E}_{q_{\mathcal{C}}(\boldsymbol{z}|\boldsymbol{x}_{\mathbb{S} \cup \mathbb{T}})} \log p_{\boldsymbol{\theta}}(\boldsymbol{x}_{\mathbb{T}}|\boldsymbol{z}, \boldsymbol{x}_{\mathbb{S}}) - \\ \beta \text{KL}[q_{\mathcal{C}}(\boldsymbol{z}|\boldsymbol{x}_{\mathbb{S} \cup \mathbb{T}})||p_{\boldsymbol{\theta}}(\boldsymbol{z}|\boldsymbol{x}_{\mathbb{S}})] \end{bmatrix}, \quad (5)$$

  where $q_{\mathcal{C}}(\boldsymbol{z}|\boldsymbol{x}_{\mathbb{S} \cup \mathbb{T}})$ is the conditional inference arm with its optimal being $q_{\mathcal{C}}^*(\boldsymbol{z}|\boldsymbol{x}_{\mathbb{S} \cup \mathbb{T}}) = p_{\boldsymbol{\theta}}(\boldsymbol{z}|\boldsymbol{x}_{\mathbb{S} \cup \mathbb{T}})$. Same with the marginal inference arm, one intuitively should parameterize different inference arms for different settings of $(\mathbb{S}, \mathbb{T})$; we will also address that issue with a universal foundation model. But different from the marginal matching inheriting analytical $p_{\boldsymbol{\theta}}(\boldsymbol{x}_{\mathbb{T}}, \boldsymbol{z})$ from the parameterized $p_{\boldsymbol{\theta}}(\boldsymbol{x}, \boldsymbol{z})$, both $p_{\boldsymbol{\theta}}(\boldsymbol{x}_{\mathbb{T}}|\boldsymbol{z}, \boldsymbol{x}_{\mathbb{S}})$ and $p_{\boldsymbol{\theta}}(\boldsymbol{z}|\boldsymbol{x}_{\mathbb{S}})$ used in the conditional matching are not readily available from the parameterized $p_{\boldsymbol{\theta}}(\boldsymbol{x}, \boldsymbol{z})$, which brings significant challenges.

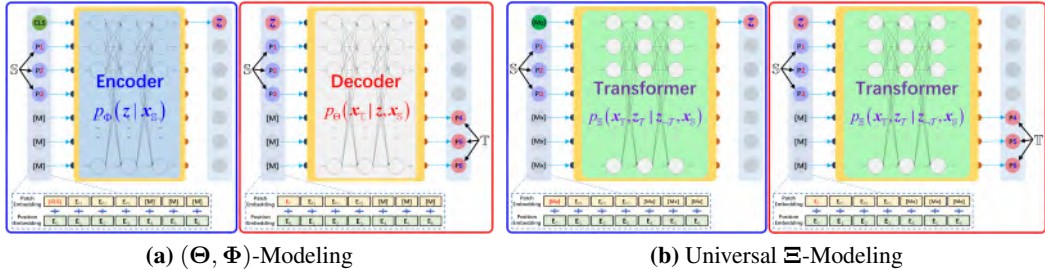

**(a)** $(\boldsymbol{\Theta}, \boldsymbol{\Phi})$-Modeling         **(b)** Universal $\boldsymbol{\Xi}$-Modeling

Figure 1: Transformer-based model architectures that are compliant to the big learning objective.

Table 1: Relationships between the proposed $(\boldsymbol{\Theta}, \boldsymbol{\Phi})$-modeling and the vanilla $(\boldsymbol{\theta}, \boldsymbol{\phi})$-modeling. The relationships are based on ideal assumptions of an *optimal* $\boldsymbol{z}$ space, where $q(\boldsymbol{x}, \boldsymbol{z})$ describes the true data generative process of $q(\boldsymbol{x})$, and $p_{\boldsymbol{\theta}^*}(\boldsymbol{x}, \boldsymbol{z}) = q(\boldsymbol{x}, \boldsymbol{z})$.

| Model | | Intermediate Process | | Ultimate Goal |
|---|---|---|---|---|
| $p_{\boldsymbol{\Theta}}(\boldsymbol{x}_{\mathbb{L}}\|\boldsymbol{z}, \boldsymbol{x}_{\emptyset})$ | $\rightarrow$ | $p_{\boldsymbol{\theta}^*}(\boldsymbol{x}\|\boldsymbol{z})$ | $\rightarrow$ | $q(\boldsymbol{x}\|\boldsymbol{z})$ |
| $p_{\boldsymbol{\Theta}}(\boldsymbol{x}_{\mathbb{T}}\|\boldsymbol{z}, \boldsymbol{x}_{\mathbb{S}})$ | $\rightarrow$ | $p_{\boldsymbol{\theta}^*}(\boldsymbol{x}_{\mathbb{T}}\|\boldsymbol{z}, \boldsymbol{x}_{\mathbb{S}})$ | $\rightarrow$ | $q(\boldsymbol{x}_{\mathbb{T}}\|\boldsymbol{z}, \boldsymbol{x}_{\mathbb{S}})$ |
| $p_{\boldsymbol{\Phi}}(\boldsymbol{z}\|\boldsymbol{x}_{\emptyset})$ | $\rightarrow$ | $p_{\boldsymbol{\theta}^*}(\boldsymbol{z})$ | $\rightarrow$ | $q(\boldsymbol{z})$ |
| $p_{\boldsymbol{\Phi}}(\boldsymbol{z}\|\boldsymbol{x}_{\mathbb{L}})$ | $\rightarrow$ | $q_{\mathcal{M}}(\boldsymbol{z}\|\boldsymbol{x}) \rightarrow p_{\boldsymbol{\theta}^*}(\boldsymbol{z}\|\boldsymbol{x})$ | $\rightarrow$ | $q(\boldsymbol{z}\|\boldsymbol{x})$ |
| $p_{\boldsymbol{\Phi}}(\boldsymbol{z}\|\boldsymbol{x}_{\mathbb{T}})$ | $\rightarrow$ | $q_{\mathcal{M}}(\boldsymbol{z}\|\boldsymbol{x}_{\mathbb{T}}) \rightarrow p_{\boldsymbol{\theta}^*}(\boldsymbol{z}\|\boldsymbol{x}_{\mathbb{T}})$ | $\rightarrow$ | $q(\boldsymbol{z}\|\boldsymbol{x}_{\mathbb{T}})$ |
| $p_{\boldsymbol{\Phi}}(\boldsymbol{z}\|\boldsymbol{x}_{\mathbb{S}\cup\mathbb{T}})$ | $\rightarrow$ | $q_{\mathcal{C}}(\boldsymbol{z}\|\boldsymbol{x}_{\mathbb{S}\cup\mathbb{T}}) \rightarrow p_{\boldsymbol{\theta}^*}(\boldsymbol{z}\|\boldsymbol{x}_{\mathbb{S}\cup\mathbb{T}})$ | $\rightarrow$ | $q(\boldsymbol{z}\|\boldsymbol{x}_{\mathbb{S}\cup\mathbb{T}})$ |

### 3.2.2 MODEL ARCHITECTURES COMPLIANT TO THE BIG LEARNING OBJECTIVE

By summarizing the above analyses associated with joint, marginal, and conditional matchings, two key modeling challenges remain unaddressed. That is,

1. **How to derive both $p_{\boldsymbol{\theta}}(\boldsymbol{x}_{\mathbb{T}}|\boldsymbol{z}, \boldsymbol{x}_{\mathbb{S}})$ and $p_{\boldsymbol{\theta}}(\boldsymbol{z}|\boldsymbol{x}_{\mathbb{S}})$ of the conditional matching from the parameterized model $p_{\boldsymbol{\theta}}(\boldsymbol{x}, \boldsymbol{z})$ of the joint matching?** Generally, this is intractable. However, we notice that

   - the marginal $p_{\boldsymbol{\theta}}(\boldsymbol{x}_{\mathbb{T}}, \boldsymbol{z})$ is readily derived from the joint $p_{\boldsymbol{\theta}}(\boldsymbol{x}, \boldsymbol{z})$ via index selection with $\mathbb{T}$; with specific $\mathbb{T} = \mathbb{L}$, the MarginELBO in (4) reduces to the JointELBO in (1);
   - the ConditionELBO in (5) is clearly more general and it recovers the JointELBO and the MarginELBO with settings of $(\mathbb{S} = \emptyset, \mathbb{T} = \mathbb{L})$ and $\mathbb{S} = \emptyset$, respectively.

   Accordingly, we propose to directly parameterize the more general $p_{\boldsymbol{\Theta}}(\boldsymbol{x}_{\mathbb{T}}|\boldsymbol{z}, \boldsymbol{x}_{\mathbb{S}})$ and $p_{\boldsymbol{\Phi}}(\boldsymbol{z}|\boldsymbol{x}_{\mathbb{S}})$ instead, where $p_{\boldsymbol{\Theta}}(\cdot)/p_{\boldsymbol{\Phi}}(\cdot)$ is constructed as a transformer-based foundation model with parameters $\boldsymbol{\Theta}/\boldsymbol{\Phi}$ (see Fig. 1a) and we use the newly introduced notations $(\boldsymbol{\Theta}, \boldsymbol{\Phi})$ to indicate a different modeling from the conventional VAE modeling with $(\boldsymbol{\theta}, \boldsymbol{\phi})$. Therefore, $p_{\boldsymbol{\Theta}}(\boldsymbol{x}, \boldsymbol{z})/p_{\boldsymbol{\Theta}}(\boldsymbol{x}_{\mathbb{T}}, \boldsymbol{z})$ can be readily retrieved for joint/marginal matching.

2. **How to concisely model the diverse inference arms $q_{\mathcal{M}}(\boldsymbol{z}|\boldsymbol{x}_{\mathbb{T}})$ and $q_{\mathcal{C}}(\boldsymbol{z}|\boldsymbol{x}_{\mathbb{S}\cup\mathbb{T}})$ w.r.t. different settings of $(\mathbb{S}, \mathbb{T})$?** We resort to the ideal situation for analysis. By considering that

   - their optima being $p_{\boldsymbol{\theta}}(\boldsymbol{z}|\boldsymbol{x}_{\mathbb{T}})$ and $p_{\boldsymbol{\theta}}(\boldsymbol{z}|\boldsymbol{x}_{\mathbb{S}\cup\mathbb{T}})$, respectively,
   - both optima have already been modeled in the parameterized $p_{\boldsymbol{\Phi}}(\boldsymbol{z}|\boldsymbol{x}_{\mathbb{S}'})$,

   we thus propose to directly leverage the parameterized $p_{\boldsymbol{\Phi}}(\boldsymbol{z}|\boldsymbol{x}_{\mathbb{T}})$ and $p_{\boldsymbol{\Phi}}(\boldsymbol{z}|\boldsymbol{x}_{\mathbb{S}\cup\mathbb{T}})$ to model those diverse inference arms w.r.t. different settings of $(\mathbb{S}, \mathbb{T})$.

The relationships between the proposed $(\boldsymbol{\Theta}, \boldsymbol{\Phi})$-modeling and the vanilla $(\boldsymbol{\theta}, \boldsymbol{\phi})$-modeling, as well as their ultimate goals, are summarized in Table 1, demonstrating the big picture of VAE modeling.

Based on the above analyses, one need two foundation models, *i.e.,* a $\boldsymbol{\Theta}$-parameterized $p_{\boldsymbol{\Theta}}(\boldsymbol{x}_{\mathbb{T}}|\boldsymbol{z}, \boldsymbol{x}_{\mathbb{S}})$ and a $\boldsymbol{\Phi}$-parameterized $p_{\boldsymbol{\Phi}}(\boldsymbol{z}|\boldsymbol{x}_{\mathbb{S}})$, both of which are capable of handling *vari-dimensional* input $\boldsymbol{x}_{\mathbb{S}}$ or output $\boldsymbol{x}_{\mathbb{T}}$. To address that issue, we borrow ideas from existing foundation models Stickland & Murray (2019); He et al. (2021) and propose to place a special mask token [M]

Table 2: Example capabilities of the big-learned universal model $p_{\boldsymbol{\Xi}}(\boldsymbol{x}_{\mathbb{T}}, \boldsymbol{z}_{\mathcal{T}}|\boldsymbol{z}_{\neg\mathcal{T}}, \boldsymbol{x}_{\mathbb{S}})$.

| Formula | Capability |
|---|---|
| $p_{\boldsymbol{\Xi}}(\boldsymbol{z}|\boldsymbol{x}), p_{\boldsymbol{\Xi}}(\boldsymbol{x}|\boldsymbol{z}), p_{\boldsymbol{\Xi}}(\boldsymbol{x}|\boldsymbol{z})p_{\boldsymbol{\Xi}}(\boldsymbol{z})$ | joint decoding, encoding, and generation |
| $p_{\boldsymbol{\Xi}}(\boldsymbol{x}_{\mathbb{T}}|\boldsymbol{z}, \boldsymbol{x}_{\mathbb{S}})$ | $\boldsymbol{x}_{\mathbb{S}}$-conditioned decoding (or $\boldsymbol{z}$-conditioned in-painting) |
| $p_{\boldsymbol{\Xi}}(\boldsymbol{z}|\boldsymbol{x}_{\mathbb{S}})$ | encoding/inference with incomplete data |
| $p_{\boldsymbol{\Xi}}(\boldsymbol{z}|\boldsymbol{x}_{\mathbb{S}_1 \cup \mathbb{S}_2})$ | encoding with combined data batches of $\boldsymbol{x}_{\mathbb{S}_1}$ and $\boldsymbol{x}_{\mathbb{S}_2}$ |
| $p_{\boldsymbol{\Xi}}(\boldsymbol{x}_{\mathbb{T}}|\boldsymbol{z}, \boldsymbol{x}_{\mathbb{S}})p_{\boldsymbol{\Xi}}(\boldsymbol{z}|\boldsymbol{x}_{\mathbb{S}})$ | arbitrary in-painting/data-completion |
| $p_{\boldsymbol{\Xi}}(\tilde{\boldsymbol{x}}_{\mathbb{T}}|\boldsymbol{z} + \Delta\boldsymbol{z}, \boldsymbol{x}_{\mathbb{S}}), p_{\boldsymbol{\Xi}}(\boldsymbol{x}_{\mathbb{T}}|\boldsymbol{z}, \boldsymbol{x}_{\mathbb{S}})$ | $\boldsymbol{x}_{\mathbb{S}}$-conditioned decoding sensitivity analysis *w.r.t.* $\Delta\boldsymbol{z}$ |
| $p_{\boldsymbol{\Xi}}(\tilde{\boldsymbol{z}}|\boldsymbol{x}_{\mathbb{S}} + \Delta\boldsymbol{x}), p_{\boldsymbol{\Xi}}(\boldsymbol{z}|\boldsymbol{x}_{\mathbb{S}})$ | encoding sensitivity analysis *w.r.t.* $\Delta\boldsymbol{x}$ |

to each location in $\mathbb{S}^{\complement}$ when inputting $\boldsymbol{x}_{\mathbb{S}}$; the vari-dimensional output $\boldsymbol{x}_{\mathbb{T}}$ is addressed via index selection.

With two flexible foundation models of $p_{\boldsymbol{\Theta}}(\boldsymbol{x}_{\mathbb{T}}|\boldsymbol{z}, \boldsymbol{x}_{\mathbb{S}})$ and $p_{\boldsymbol{\Phi}}(\boldsymbol{z}|\boldsymbol{x}_{\mathbb{S}})$, we are ready to finalize the BigLearn-VAE. Before that, we go one step further on designing the model architecture. Specifically, we observe in Table 1 that both $p_{\boldsymbol{\Theta}}(\boldsymbol{x}_{\mathbb{T}}|\boldsymbol{z}, \boldsymbol{x}_{\mathbb{S}})$ and $p_{\boldsymbol{\Phi}}(\boldsymbol{z}|\boldsymbol{x}_{\mathbb{S}})$ have ultimate goals associated with (different perspectives of) the unique $p_{\boldsymbol{\theta}^*}(\boldsymbol{x}, \boldsymbol{z}) = q(\boldsymbol{x}, \boldsymbol{z})$; such consistency on their ultimate goals is akin to what motivated the universal modeling of the big learning Cong & Zhao (2022). Accordingly, we propose to further unify $\boldsymbol{\Theta}$ and $\boldsymbol{\Phi}$ by constructing a $\boldsymbol{\Xi}$-parameterized universal foundation model $p_{\boldsymbol{\Xi}}(\boldsymbol{x}_{\mathbb{T}}, \boldsymbol{z}_{\mathcal{T}}|\boldsymbol{z}_{\neg\mathcal{T}}, \boldsymbol{x}_{\mathbb{S}})$, where the bool $\mathcal{T} = $ True/False indicates $\boldsymbol{z}$ is in the output/input. With different settings for $(\mathbb{S}, \mathbb{T}, \mathcal{T})$, the universal model $p_{\boldsymbol{\Xi}}(\boldsymbol{x}_{\mathbb{T}}, \boldsymbol{z}_{\mathcal{T}}|\boldsymbol{z}_{\neg\mathcal{T}}, \boldsymbol{x}_{\mathbb{S}})$ is capable of modeling both the diverse encoding $p_{\boldsymbol{\Phi}}(\boldsymbol{z}|\boldsymbol{x}_{\mathbb{S}})$ and the diverse decoding $p_{\boldsymbol{\Theta}}(\boldsymbol{x}_{\mathbb{T}}|\boldsymbol{z}, \boldsymbol{x}_{\mathbb{S}})$; see Fig. 1b for explicit demonstrations.

### 3.2.3 Finalizing the BigLearn-VAE

With two flexible foundation models of $p_{\boldsymbol{\Theta}}(\boldsymbol{x}_{\mathbb{T}}|\boldsymbol{z}, \boldsymbol{x}_{\mathbb{S}})$ and $p_{\boldsymbol{\Phi}}(\boldsymbol{z}|\boldsymbol{x}_{\mathbb{S}})$ and the key idea of exhaustive exploitation of the data information via the big learning, *i.e.,* simultaneous joint, marginal, and conditional matchings, we finalize the tailored big learning objective for the BigLearn-VAE with *two* models as

$$\text{BigLearnELBO}_T(\boldsymbol{\Theta}, \boldsymbol{\Phi}) = \mathbb{E}_{q(\mathbb{S}, \mathbb{T})q(\boldsymbol{x}_{\mathbb{S} \cup \mathbb{T}})} \left[ \begin{array}{l} \mathbb{E}_{p_{\boldsymbol{\Phi}}(\boldsymbol{z}|\boldsymbol{x}_{\mathbb{S} \cup \mathbb{T}})} \log p_{\boldsymbol{\Theta}}(\boldsymbol{x}_{\mathbb{T}}|\boldsymbol{z}, \boldsymbol{x}_{\mathbb{S}})- \\ \beta \text{KL}[p_{\boldsymbol{\Phi}}(\boldsymbol{z}|\boldsymbol{x}_{\mathbb{S} \cup \mathbb{T}})||p_{\boldsymbol{\Phi}}(\boldsymbol{z}|\boldsymbol{x}_{\mathbb{S}})] \end{array} \right], \qquad (6)$$

where $q(\mathbb{S}, \mathbb{T})$ denotes the sampling process the $(\mathbb{S}, \mathbb{T})$ pair and it implicitly defines the weighting among joint, marginal, and conditional matching tasks. With different settings for $q(\mathbb{S}, \mathbb{T})$, the BigLearnELBO can recover the JointELBO in (1), the MarginELBO in (4), and the ConditionELBO in (5).

Similarly, the tailored big learning objective for the BigLearn-VAE with *one* universal model $p_{\boldsymbol{\Xi}}(\boldsymbol{x}_{\mathbb{T}}, \boldsymbol{z}_{\mathcal{T}}|\boldsymbol{z}_{\neg\mathcal{T}}, \boldsymbol{x}_{\mathbb{S}})$ is defined as

$$\text{BigLearnELBO}_U(\boldsymbol{\Xi}) = \mathbb{E}_{q(\mathbb{S}, \mathbb{T})q(\boldsymbol{x}_{\mathbb{S} \cup \mathbb{T}})} \left[ \begin{array}{l} \mathbb{E}_{p_{\boldsymbol{\Xi}}(\boldsymbol{z}|\boldsymbol{x}_{\mathbb{S} \cup \mathbb{T}})} \log p_{\boldsymbol{\Xi}}(\boldsymbol{x}_{\mathbb{T}}|\boldsymbol{z}, \boldsymbol{x}_{\mathbb{S}})- \\ \beta \text{KL}[p_{\boldsymbol{\Xi}}(\boldsymbol{z}|\boldsymbol{x}_{\mathbb{S} \cup \mathbb{T}})||p_{\boldsymbol{\Xi}}(\boldsymbol{z}|\boldsymbol{x}_{\mathbb{S}})] \end{array} \right], \qquad (7)$$

with which we harvest from *one* universal foundation model a BigLearn-VAE. After the big learning, that universal model is expected to possess various data capabilities simultaneously, which is likely valuable for versatile data analysis and manipulation. See Table 2 for example capabilities.

## 4 Related Work

**VAE Variants.** Plenty of improved variants of the variational auto-encoder (VAE) have been developed since its proposal Kingma & Welling (2014); Rezende et al. (2014), with efforts made towards ($i$) a better modeling of the $\boldsymbol{x}$-manifold Dai & Wipf (2019a), ($ii$) a better $\boldsymbol{z}$-prior Tomczak & Welling (2018); Casale et al. (2018); Davidson et al. (2018); Takahashi et al. (2019); Joo et al. (2020), ($iii$) a better balance between the reconstruction and KL terms Higgins et al. (2016); Chen et al. (2018); Castrejon et al. (2019); Bae et al. (2022), ($iv$) a more powerful inference arm Rezende & Mohamed (2015); Kingma et al. (2016), and ($v$) a better training objective Burda et al. (2015);

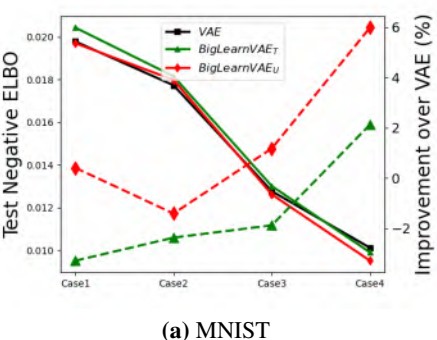 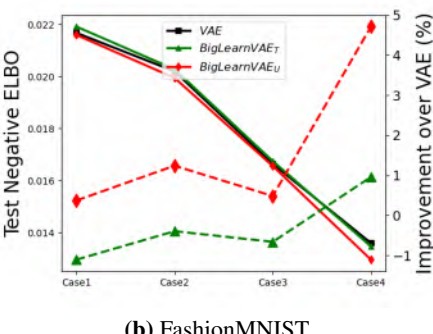

**(a)** MNIST                                    **(b)** FashionMNIST

Figure 2: BigLearn-VAE versus the conventional jointly-trained VAE. Note the test JointELBO is selected as the evaluation metric. From Case1 to Case4, increasing number of trainable model parameters are utilized.

Tolstikhin et al. (2017); Zhao et al. (2019); Hao & Shafto (2023); Estermann & Wattenhofer (2023). Different from existing methods, the proposed BigLearn-VAE upgrade the conventional VAE from a new big-learning dimension that is inspired by ground-breaking foundation models. Besides the vanilla encoding-decoding architectures are also upgraded to enable carrying diverse data capabilities.

**Conditional Sampling With Jointly-Trained VAEs.** Conditional sampling is a key challenge for downstream applications of VAEs Rezende et al. (2014); Nguyen et al. (2017); Duan et al. (2019); Harvey et al. (2021); Simkus & Gutmann (2023). To harvest the conditional-sampling capability from a jointly-trained VAE, existing methods leverage post-processing techniques, such as Gibbs sampling that reuses the encoder Rezende et al. (2014); Mattei & Frellsen (2018), Markov chain Monte Carlo (MCMC) Wu et al. (2018), and variational inference Nguyen et al. (2017); Harvey et al. (2021). However, the reliability of such post-processed conditional-sampling capability highly depends on the performance of the original jointly-trained VAE; besides, pitfalls likely exist Simkus & Gutmann (2023). By comparison, the presented BigLearn-VAE utilize the conditional sampling (*i.e.,* conditional matching) as one of its training tasks, leading to *explicitly* trained conditional-sampling capability on diverse conditional data samples.

**Foundation Models & Big Learning.** AI is undergoing a paradigm shift with the rise of foundation models Bommasani et al. (2021); Yuan et al. (2022), such as the popular BERT (Stickland & Murray, 2019), GPTs (Brown et al., 2020; Ouyang et al., 2022; OpenAI, 2022; 2023), the MAE (He et al., 2021), DALL-Es (Ramesh et al., 2021; 2022), *etc.* Foundation models are well known to succeed from its large-scale pretraining on broad data at scale; however, less attention has been paid to the underlying principle of its pretraining objectives (Bommasani et al., 2021; Yuan et al., 2022), except for the big learning Cong & Zhao (2022). Different from the main research stream of foundation models that pursues massive data and huge models, we focus on utilizing their underlying learning principle, *i.e.,* the big learning, to upgrade traditional machine learning paradigm of VAEs.

## 5 EXPERIMENTS

To demonstrate the effectiveness of the presented BigLearn-VAE, we first quantitatively compare it with the conventional jointly-trained VAE; then, we qualitatively illustrate the big-learned data capabilities; and finally, we reveal that the BigLearn-VAE encoder can serve as a reliable feature extractor for down-streaming incomplete-data classifications.

### 5.1 BIGLEARN-VAE VERSUS JOINTLY-TRAINED VAE

We quantitatively compare the presented BigLearn-VAE with the conventional jointly-trained VAE, utilizing the test JointELBO that favors the jointly-trained VAE as the evaluation metric. Note this is a challenging and even unfair setup for the BigLearn-VAE, because the big-learning objective consists of plenty of training tasks and the joint matching is merely one of them.

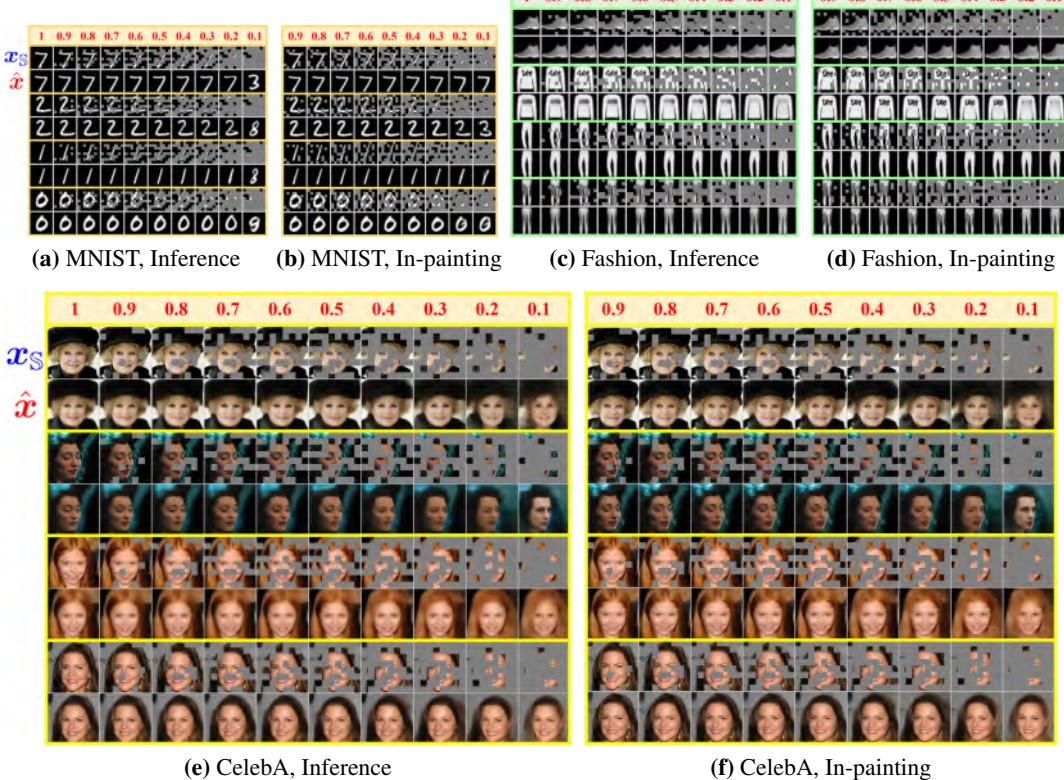

**(a)** MNIST, Inference    **(b)** MNIST, In-painting    **(c)** Fashion, Inference    **(d)** Fashion, In-painting

**(e)** CelebA, Inference

**(f)** CelebA, In-painting

Figure 3: Demonstration of the big-learned capabilities of inference with incomplete data (a, c, e) and arbitrary in-painting (b, d, f). For inference with incomplete data, the inferred latent code $\hat{z} \sim p_{\Xi}(z|x_{\mathbb{S}})$ and its representativeness is shown via the reconstruction $\hat{x} \sim p_{\Xi}(x|\hat{z})$. $x_{\mathbb{S}}$ is shown in the first row with the $\mathbb{S}$-ratio decreasing from 1 to 0.1 (from the left to the right). Arbitrary in-painting in (b, d, f) is implemented with $\hat{x}_{\mathbb{T}} \sim p_{\Xi}(x_{\mathbb{T}}|\hat{z}, x_{\mathbb{S}}), \hat{z} \sim p_{\Xi}(z|x_{\mathbb{S}})$. More demonstrations are given in Appendix C.

Experiments are conducted on the benchmark MNIST and FashionMNIST datasets. Three VAEs are compared, *i.e.,* the conventional VAE jointly trained with (1) (marked as VAE), the BigLearn-VAE with the $(\Theta, \Phi)$-modeling and the objective in (6) (BigLearnVAE$_T$), and the BigLearn-VAE with the universal $\Xi$-modeling and the objective in (7) (BigLearnVAE$_U$). Four different model settings are considered, where the number of trainable parameters are 1.2103M (marked as Case1), 4.733M (Case2), 4.78M (Case3), and 23.0081M (Case4), respectively. For BigLearnVAE$_U$ with a universal model, we deepen the network architecture to keep the same trainable parameters for fair comparisons. See Appendix A for the detailed model architectures and other experimental settings.

Fig. 2 demonstrates the experimental results. It's clear that, ($i$) when compared with VAE, both BigLearnVAE$_T$ and BigLearnVAE$_U$ delivers overall comparable or better test JointELBO, justifying the effectiveness of the big learning; ($ii$) BigLearnVAE$_U$ consistently outperforms BigLearnVAE$_T$ as expected, which highlights the benefit of the universal $\Xi$-modeling (that is akin to the big learning principle); and ($iii$) as the model size increases, both BigLearn-VAEs demonstrate overall increasing improvements over the jointly-trained VAE; this is expected because a larger model capacity will better conform to the massive training nature of the big learning and diverse joint, marginal, and conditional matchings may work better in encouraging model parameters to concentrate on the data essence.

## 5.2 DEMONSTRATION OF THE BIG-LEARNED DATA CAPABILITIES

Below we qualitatively demonstrate the big-learned data capabilities, with a main focus on those associated with incomplete data. Specifically, we test the big-learned capabilities associated with ($i$) inference with incomplete data and ($ii$) arbitrary in-painting/data-completion.

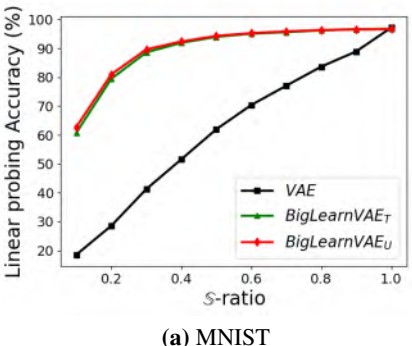 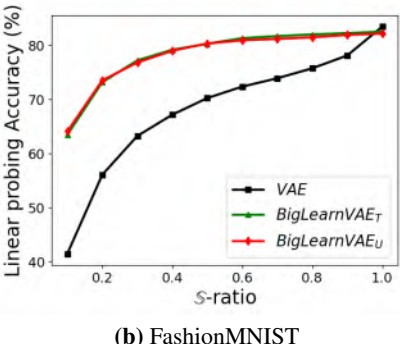

**(a)** MNIST         **(b)** FashionMNIST

Figure 4: Big-learned latent codes are robust to down-streaming incomplete-data classifications.

Fig. 3 demonstrate the corresponding results; those on MNIST and FashionMNIST are from the Case4 BigLearnVAE$_U$ models, while those on CelebA are from a larger BigLearnVAE$_U$ model (see Appendix B for details).

From Figs. 3a, 3c, and 3e, it's clear that the inferred latent codes $\hat{z} \sim p_{\Xi}(z|x_{\mathbb{S}})$ are quite robust towards data incompleteness, as they stably generate the correct/similar digits/products/faces even when extracted with a $\mathbb{S}$-ratio of $0.2/0.1/0.2$. Note when the $\mathbb{S}$-ratio equals $0.1$, the input $x_{\mathbb{S}}$ in the MNIST experiment may not have digit information. Figs. 3b, 3d, and 3f show the results for arbitrary in-painting, *i.e.,* $\hat{x}_{\mathbb{T}} \sim p_{\Xi}(x_{\mathbb{T}}|\hat{z}, x_{\mathbb{S}}), \hat{z} \sim p_{\Xi}(z|x_{\mathbb{S}})$; it's clear that the BigLearn-VAE delivers overall realistic in-paintings, highlighting again the effectiveness of the big learning. By parallel comparing the inference results with those from in-paintings, we observe that $(i)$ the extracted latent code $\hat{z}$ plays a dominated role in mastering the primary information associated with digit/product classes or the overall photo scene, whereas $(ii)$ the input $x_{\mathbb{S}}$ of $\hat{x}_{\mathbb{T}} \sim p_{\Xi}(x_{\mathbb{T}}|\hat{z}, x_{\mathbb{S}})$ brings to the decoded images the detailed information, such as the facial details that reflects the identity.

### 5.3 BIG-LEARNED LATENT CODES ARE ROBUST TO INCOMPLETE-DATA CLASSIFICATIONS

Noticing the dominated role played by the latent code $\hat{z}$ in Fig. 3, we conduct additional experiments to test whether the big-learned latent codes are robust to incomplete-data classifications.

Specifically, we randomly mask the MNIST and FashionMNIST datasets with different $\mathbb{S}$-ratios to mimic incomplete-data classification scenarios with various degrees of data incompleteness. We then utilize the big-learned $p_{\Xi}(z|x_{\mathbb{S}})$ as a pretrained feature extractor that is amenable to incomplete data. Finally, we follow He et al. (2021) to employ linear probing on top of the extracted codes and use the linear-probing accuracy to evaluate the robustness of the big-learned latent codes *w.r.t.* incomplete-data classifications.

The experimental results are summarized in Fig. 4, where it's clear that BigLearn-VAEs delivers robust latent-codes/features for down-streaming incomplete-data classifications. It's worth highlighting that, even with severe data incompleteness with merely $10\%$ observed patches (*i.e.,* $\mathbb{S}$-ratio being $0.1$), the big-learned feature extractor $p_{\Xi}(z|x_{\mathbb{S}})$ delivers $\geq 60\%$ linear-probing accuracy.

## 6 CONCLUSIONS

We leverage the big learning principle Cong & Zhao (2022) to upgrade the conventional jointly-trained VAE into the BigLearn-VAE, which delivers versatile and valuable data capabilities (like conditional sampling) with one universal foundation model and with comparable/better joint performance. Experimental results demonstrate its effectiveness. Further research might extend the big-learning principle to the $(x, z)$-space or conduct diverse data analysis based on the BigLearn-VAE.

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

# Appendix of Big Learning Variational Auto-Encoders

**Anonymous Authors**

## A   EXPERIMENTAL SETTINGS ON MNIST AND FASHIONMNIST

Table 3: Model settings on the MNIST and FashionMNIST datasets. The transformer/ViT architecture is employed for all the listed experiments. Regarding the "# Layer" column, x-y indicates utilizing a x-layer encoder and y-layer decoder; for BigLearnVAE$_U$ with a universal foundation model, "# Layer" indicates the number of its transformer layers.

| Case | Method | # Layer | $z$-Dimension | Embedding Dimension | # Trainable Parameters |
|------|--------|---------|---------------|---------------------|------------------------|
|       | VAE           | 4-4 | 32  | 128 | 1.2103M |
| Case1 | BigLearnVAE$_T$ | 4-4 | 32  | 128 | 1.2103M |
|       | BigLearnVAE$_U$ | 8   | 32  | 128 | 1.2103M |
|       | VAE           | 4-4 | 32  | 256 | 4.733M |
| Case2 | BigLearnVAE$_T$ | 4-4 | 32  | 256 | 4.733M |
|       | BigLearnVAE$_U$ | 8   | 32  | 256 | 4.733M |
|       | VAE           | 4-4 | 64  | 256 | 4.78M |
| Case3 | BigLearnVAE$_T$ | 4-4 | 64  | 256 | 4.78M |
|       | BigLearnVAE$_U$ | 8   | 64  | 256 | 4.78M |
|       | VAE           | 6-4 | 128 | 512 | 23.0081M |
| Case4 | BigLearnVAE$_T$ | 6-4 | 128 | 512 | 23.0081M |
|       | BigLearnVAE$_U$ | 10  | 128 | 512 | 23.0081M |

Table 3 summarizes the detailed model settings on the MNIST and FashionMNIST datasets. All the input images are resized to $32 \times 32$. The patch size is set as $4$, leading to dimension $D = 48$ and length $L = 64$. $\beta = 0.01$ by default. The AdamW Loshchilov & Hutter (2017) optimizer with $\beta_1 = 0$, $\beta_2 = 0.999$, and $\epsilon = 1 \times 10^{-8}$ is used as the default optimizer. The learning rate is set as $1 \times 10^{-4}$ for Case1, Case2, and Case3 and $2 \times 10^{-5}$ for Case4.

## B   IMPLEMENTATION DETAILS ON CELEBA

Table 4: Model settings on the CelebA dataset. A convolutional neural network with residual is used to construct the PatchAE in Stage1. A base transformer/ViT is employed in Stage2 to construct the universal foundation model of the BigLearn-VAE.

|  | # Layer | $z$-Dimension | Embedding Dimension | # Trainable Parameters |
|--|---------|---------------|---------------------|------------------------|
| PatchAE (Stage1) | - | 512 | - | 2.3193M |
| BigLearn-VAE (Stage2) | 12 | 128 | 768 | 89.9421M |

Table 3 summarizes the model settings on the CelebA dataset. All the input images are resized to $128 \times 128$. The patch size is set as $16$, leading to dimension $D = 768$ and length $L = 64$. $\beta = 0.001$ by default. The AdamW optimizer with $\beta_1 = 0.9$, $\beta_2 = 0.95$, and $\epsilon = 1 \times 10^{-5}$ is used. The learning rate is set as $1 \times 10^{-4}$ and a cosine decay learning rate scheduler is adopted.

Fig. 5 demonstrates the big picture of training a BigLearn-VAE on the CelebA dataset. We follow Dai & Wipf (2019b) to employ a two-stage training strategy, as detailed below.

- In Stage1, we train a patch-level auto-encoder (Patch-AE) to embed the CelebA manifold into a low-dimensional transformed space, where the goal is to bypass the manifold modeling challenge and, at the same time, to figure out a transformed space where the transformed data are easy to model for Stage2,

- Based on the Patch-AE that is frozen from Stage1, we then conduct the BigLearn-VAE in Stage2 to perform big learning in the transformed space. Note the pipeline consisting of a patch-level AE and a follow-up BigLearn-VAE will not prevent collecting versatile data capabilities like conditional sampling.

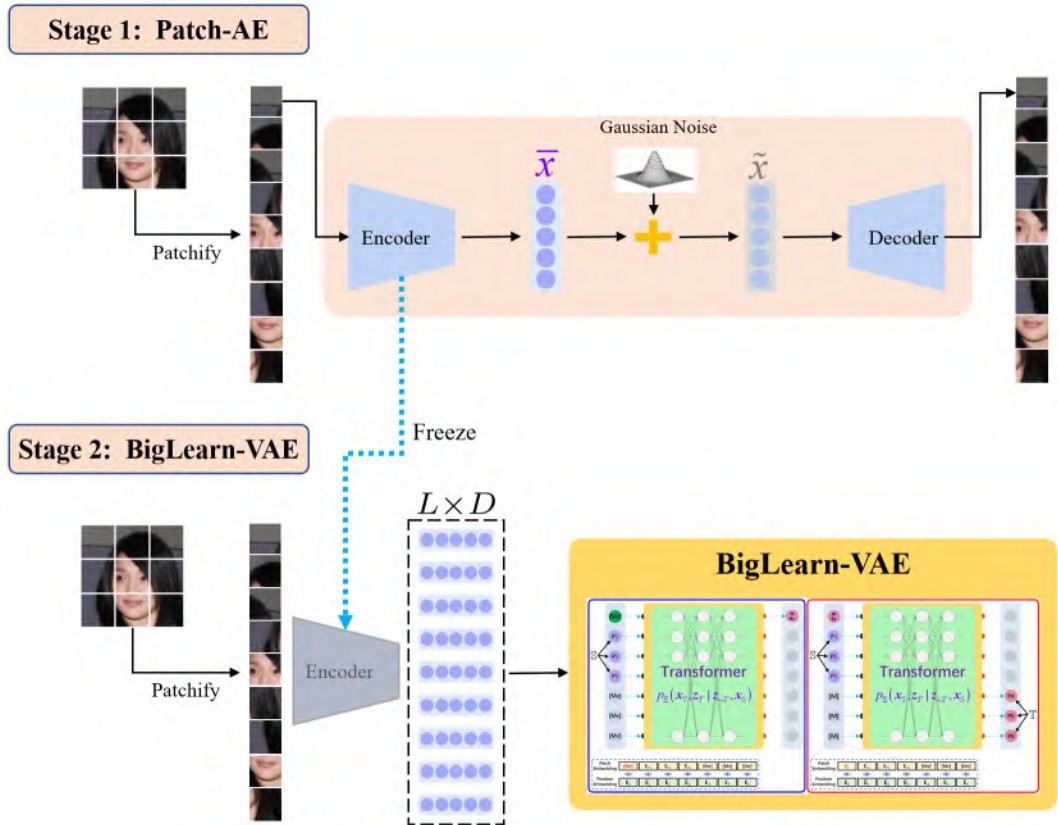

Figure 5: The big picture of training a BigLearn-VAE on the CelebA.

## C    MORE DEMONSTRATIONS

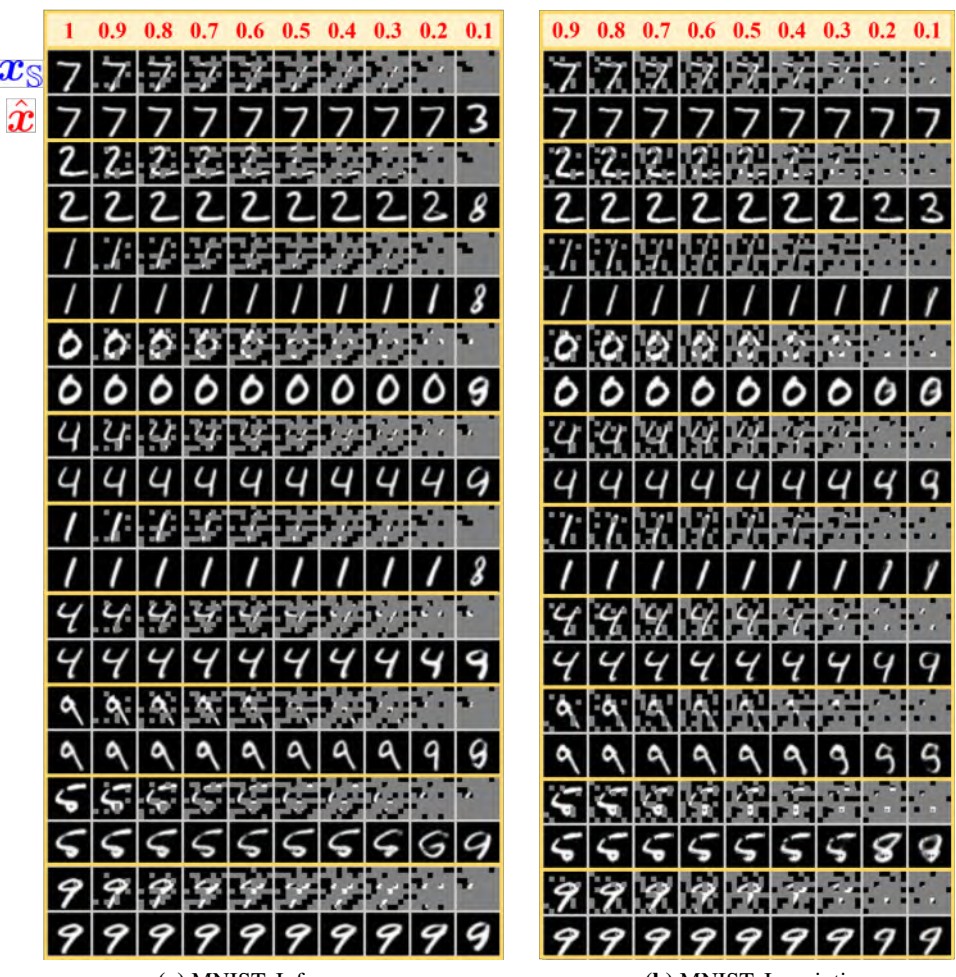

(a) MNIST, Inference        (b) MNIST, In-painting

Figure 6: Demonstration of the big-learned capabilities of inference with incomplete data (a) and arbitrary in-painting (b) on MNIST.

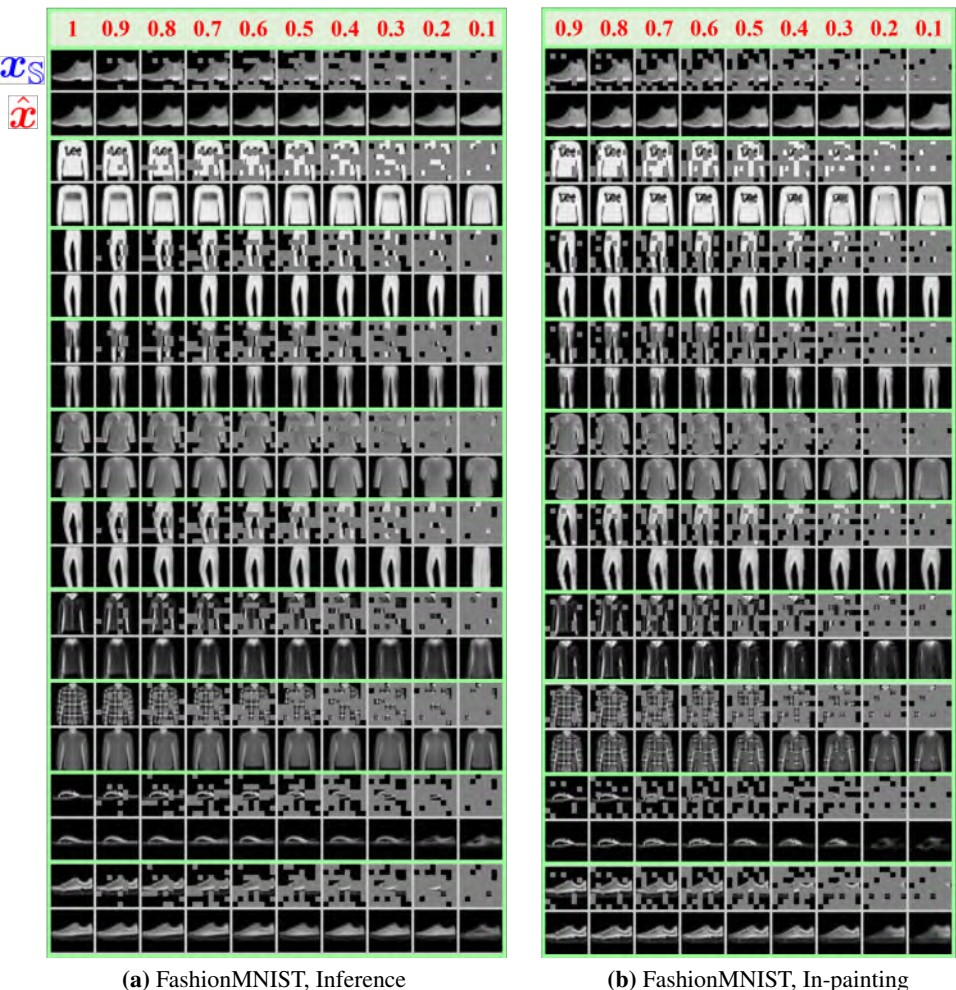

**(a)** FashionMNIST, Inference
**(b)** FashionMNIST, In-painting

Figure 7: Demonstration of the big-learned capabilities of inference with incomplete data (a) and arbitrary in-painting (b) on FashionMNIST.

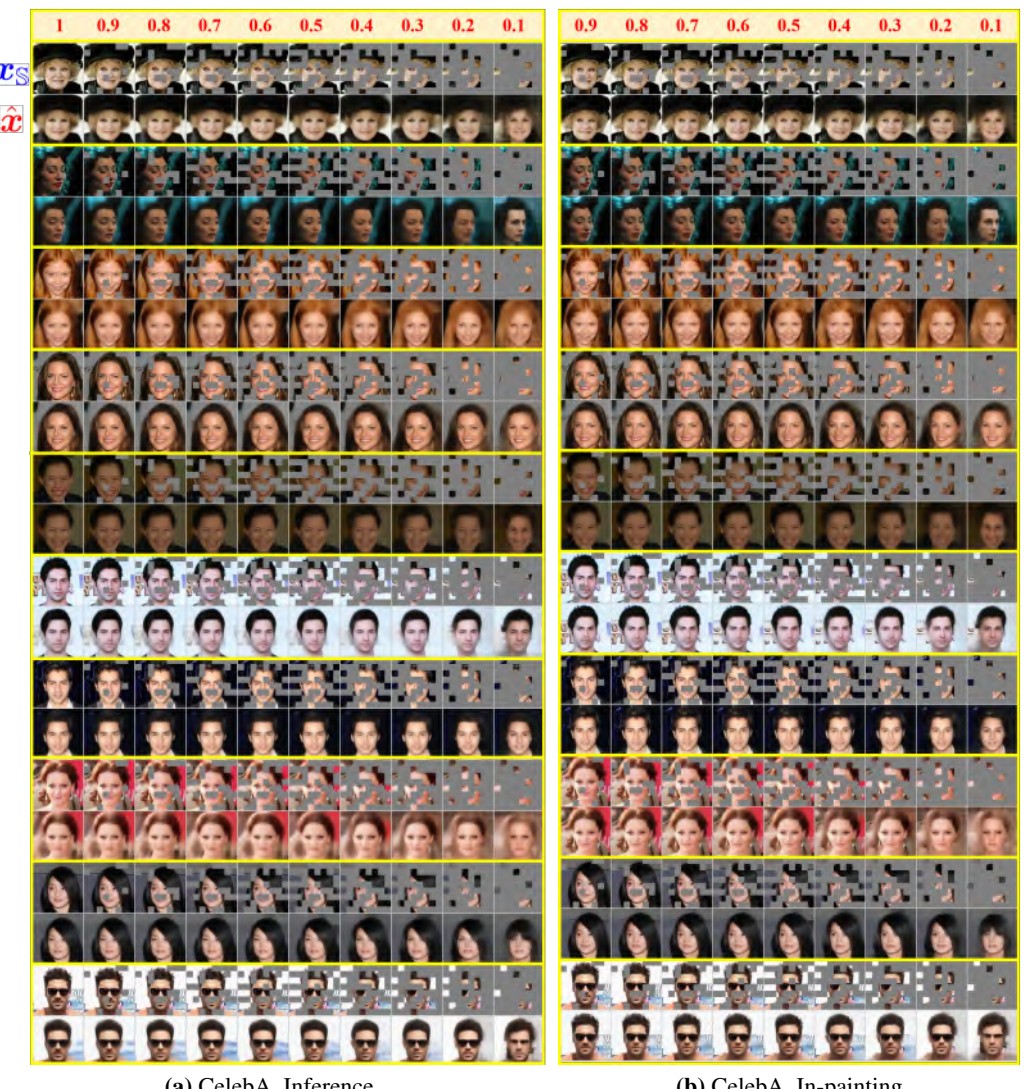

(a) CelebA, Inference    (b) CelebA, In-painting

Figure 8: Demonstration of the big-learned capabilities of inference with incomplete data (a) and arbitrary in-painting (b) on CelebA.

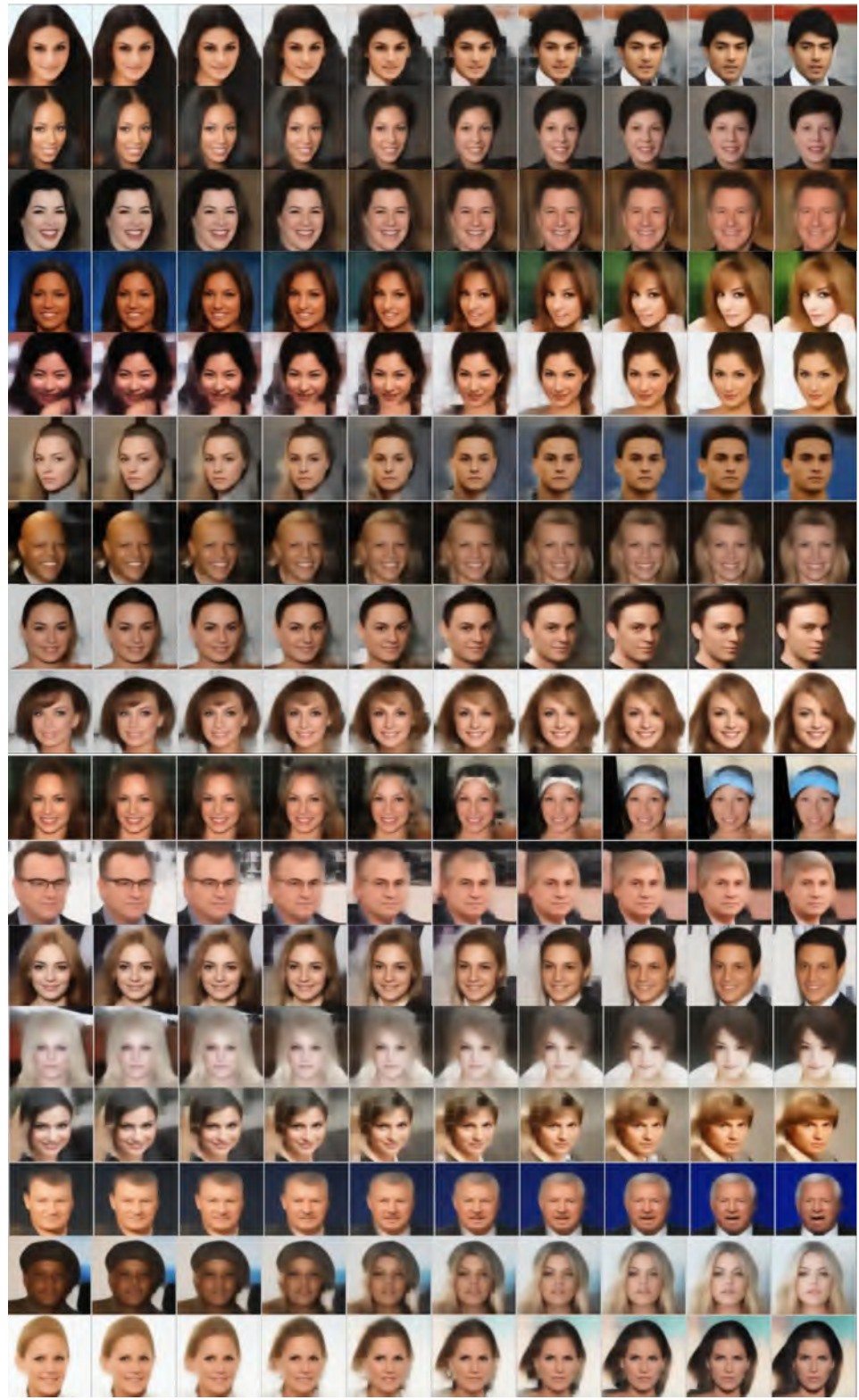

Figure 9: Linear interpolations on the latent codes from the BigLearn-VAE trained on the CelebA dataset.

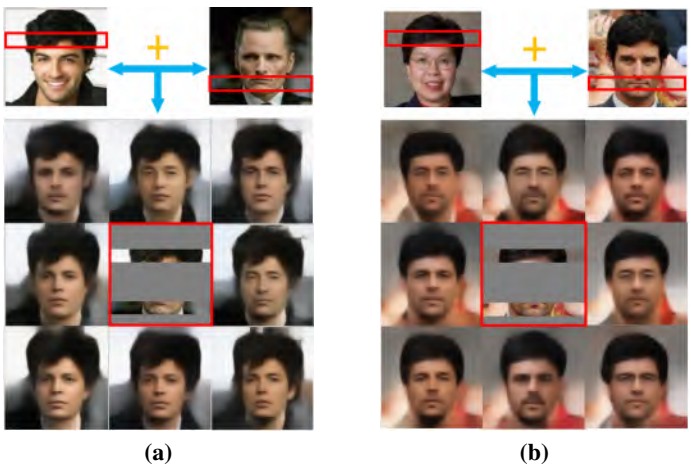

**(a)**          **(b)**

Figure 10: Test the big-learned data capability of encoding/inference with combined data batches of $x_{\mathbb{S}_1}$ and $x_{\mathbb{S}_2}$, *i.e.*, $\hat{z} \sim p_{\Xi}(z|x_{\mathbb{S}_1 \cup \mathbb{S}_2})$. The representativeness of the latent code $\hat{z}$ is illustrated via decoding, *i.e.*, $\hat{x}_{\mathbb{T}} \sim p_{\Xi}(x_{\mathbb{T}}|\hat{z}, x_{\mathbb{S}})$.

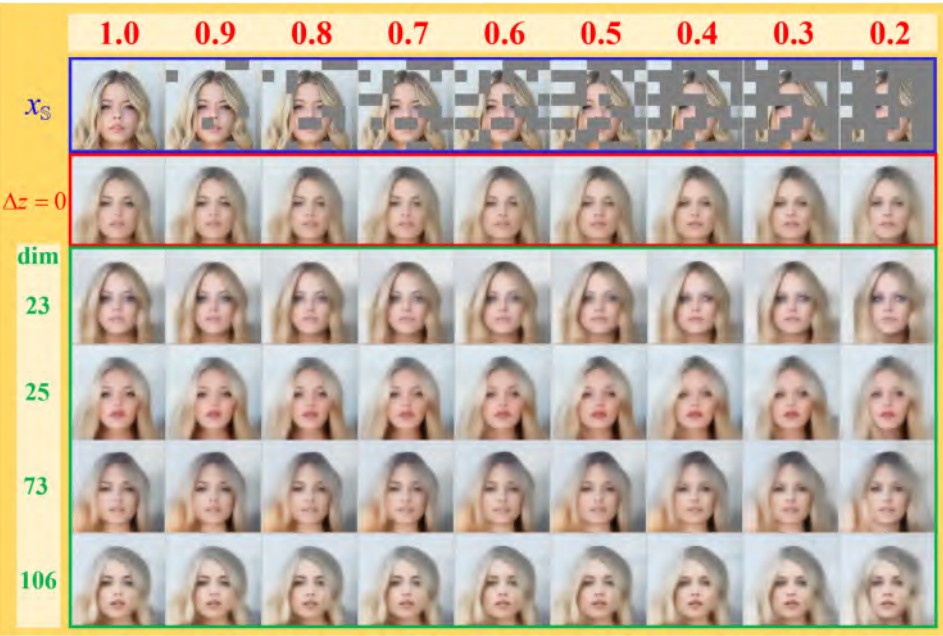

Figure 11: Test the big-learned data capability of $x_{\mathbb{S}}$-conditioned decoding sensitivity analysis *w.r.t.* $\Delta z$, *i.e.*, $p_{\Xi}(\bar{x}_{\mathbb{T}}|z + \Delta z, x_{\mathbb{S}}), p_{\Xi}(x_{\mathbb{T}}|z, x_{\mathbb{S}})$. $\Delta z = 5$ by default. It seems that $z_{23}$ (*i.e.*, the 23-th element of $z$), $z_{25}$, $z_{73}$, and $z_{106}$ control eye make-up, lipstick, hair color, and hair style respectively. More interestingly, that control is consistent *w.r.t.* different degrees of data incompleteness, highlighting the robustness of the big-learned latent codes.

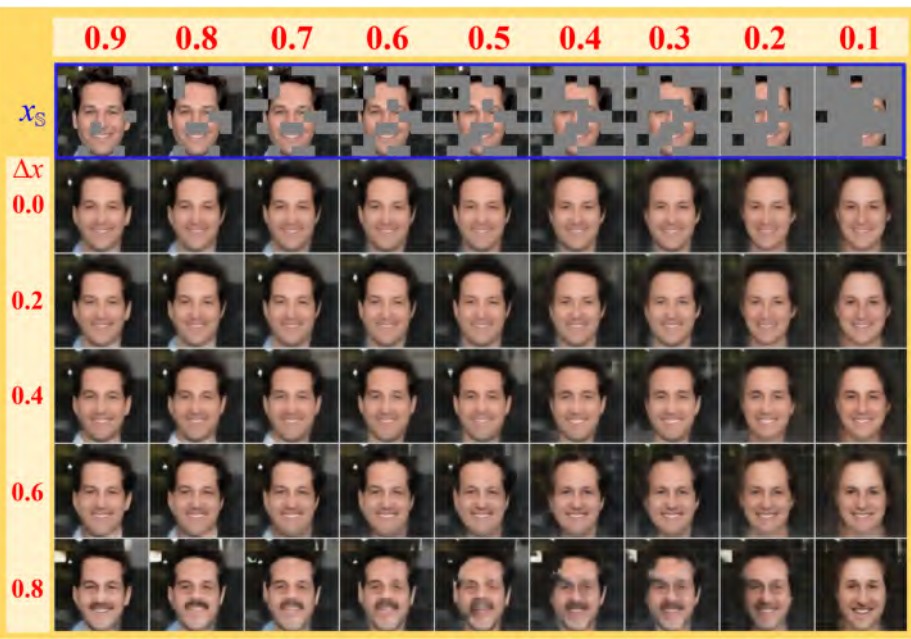

Figure 12: Test the big-learned data capability of encoding sensitivity analysis *w.r.t.* $\Delta \boldsymbol{x}$, *i.e.*, $p_{\Xi}(\bar{\boldsymbol{z}}|\boldsymbol{x}_{\mathbb{S}} + \Delta \boldsymbol{x}), p_{\Xi}(\boldsymbol{z}|\boldsymbol{x}_{\mathbb{S}})$. Both $\bar{\boldsymbol{z}}$ and $\boldsymbol{z}$ are illustrated via decoding, *i.e.,* $\bar{\boldsymbol{x}}_{\mathbb{T}} \sim p_{\Xi}(\boldsymbol{x}_{\mathbb{T}}|\bar{\boldsymbol{z}}, \boldsymbol{x}_{\mathbb{S}})$ and $\boldsymbol{x}_{\mathbb{T}} \sim p_{\Xi}(\boldsymbol{x}_{\mathbb{T}}|\boldsymbol{z}, \boldsymbol{x}_{\mathbb{S}})$, respectively.

## D  ALGORITHM FLOW

---
**Algorithm 1** Big Learning Variational Auto-Encoders

---
**Input:** Training data $\mathbf{X}$, # of steps $S$.
**Output:** Predicted image patches $\mathbf{x}_{\mathbb{T}}$.
 1:  $\Phi \leftarrow$ Initialize parameters
 2:  Random minibatch $\mathbf{x}$ of real data
 3:  **for** step$\in \{1 \dots S\}$ **do**
 4:      S-ratio $\sim$ Beta$(\alpha_1, \beta_1)$ and T-ratio $\sim$ Beta$(\alpha_2, \beta_2)$
 5:      uniformly sample an index subset $\mathbb{S}$ and $\mathbb{T}$
 6:      Encode: $\mathbf{z} \sim p_\Phi \left( \mathbf{z} \mid \boldsymbol{x}_{\mathbb{S}\cup\mathbb{T}} \right)$
 7:      Decode: $\mathbf{x}_{\mathbb{T}} \sim p_\Phi \left( \boldsymbol{x}_{\mathbb{T}} \mid \mathbf{z}, \boldsymbol{x}_{\mathbb{S}} \right)$
 8:      Compute BigLearnELBO as defined in manuscript
 9:      Update $\Phi$
10:  **end for**

---

## E  TRAINING TIME

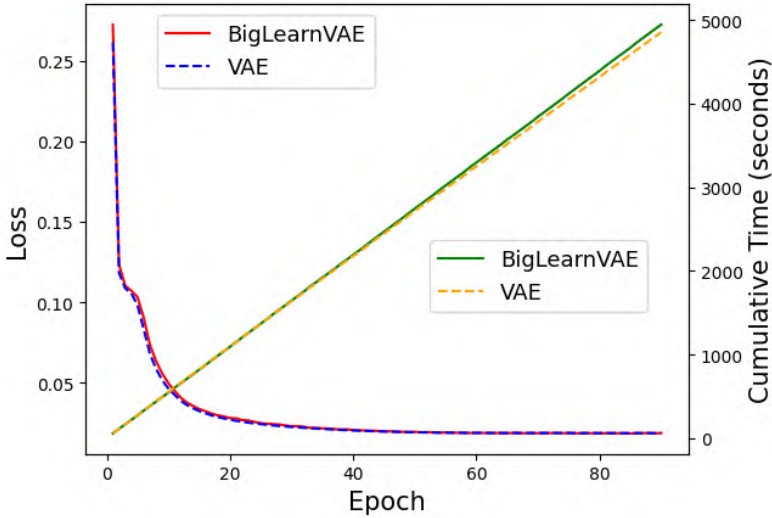

Figure 13: Under the same settings, the Loss and Cumulative Time per Epoch of jointly-trained VAE and BigLearnVAE.

## F  EXAMPLES OF MODEL CAPABILITIES

Fig.14 showcases the diverse capabilities of BigLearnVAE, including Sample/Generation, Inference, Reconstruction, In-painting and Conditional Sample/Generation. Specifically, Sample/Generation involves sampling from the prior to obtain the latent variable $\boldsymbol{z}$, followed by generating images through the decoder $p(\boldsymbol{x}|\boldsymbol{z})$. Inference refers to the model's need to deduce missing image areas based on available image regions, where the encoder obtains the latent variable $\boldsymbol{z}$ through $p(\boldsymbol{z}|\boldsymbol{x}_{part})$, and then the missing image areas are inferred through the decoder $p(\boldsymbol{x}_{miss}|\boldsymbol{z})$. Reconstruction operates similarly to traditional VAE, where the encoder acquires the latent variable $\boldsymbol{z}$ through $p(\boldsymbol{z}|\boldsymbol{x}_{complete})$, and then reconstructs images through the decoder $p(\boldsymbol{x}_{rec}|\boldsymbol{z})$. In-painting, similar to Inference, generates predicted image areas based on existing image regions, but differs in that the decoding stage can be conditioned on $\boldsymbol{x}_s$, which is $p(\boldsymbol{x}_{miss}|\boldsymbol{z}, \boldsymbol{x}_s)$. Conditional Sample/Generation differs from Sample/Generation in that the encoding stage can incorporate partial information to guide the direction of generation, as shown in Fig.14e, where given partial stroke

information, BigLearnVAE can generate digit images that meet the conditions and possess a certain degree of diversity.

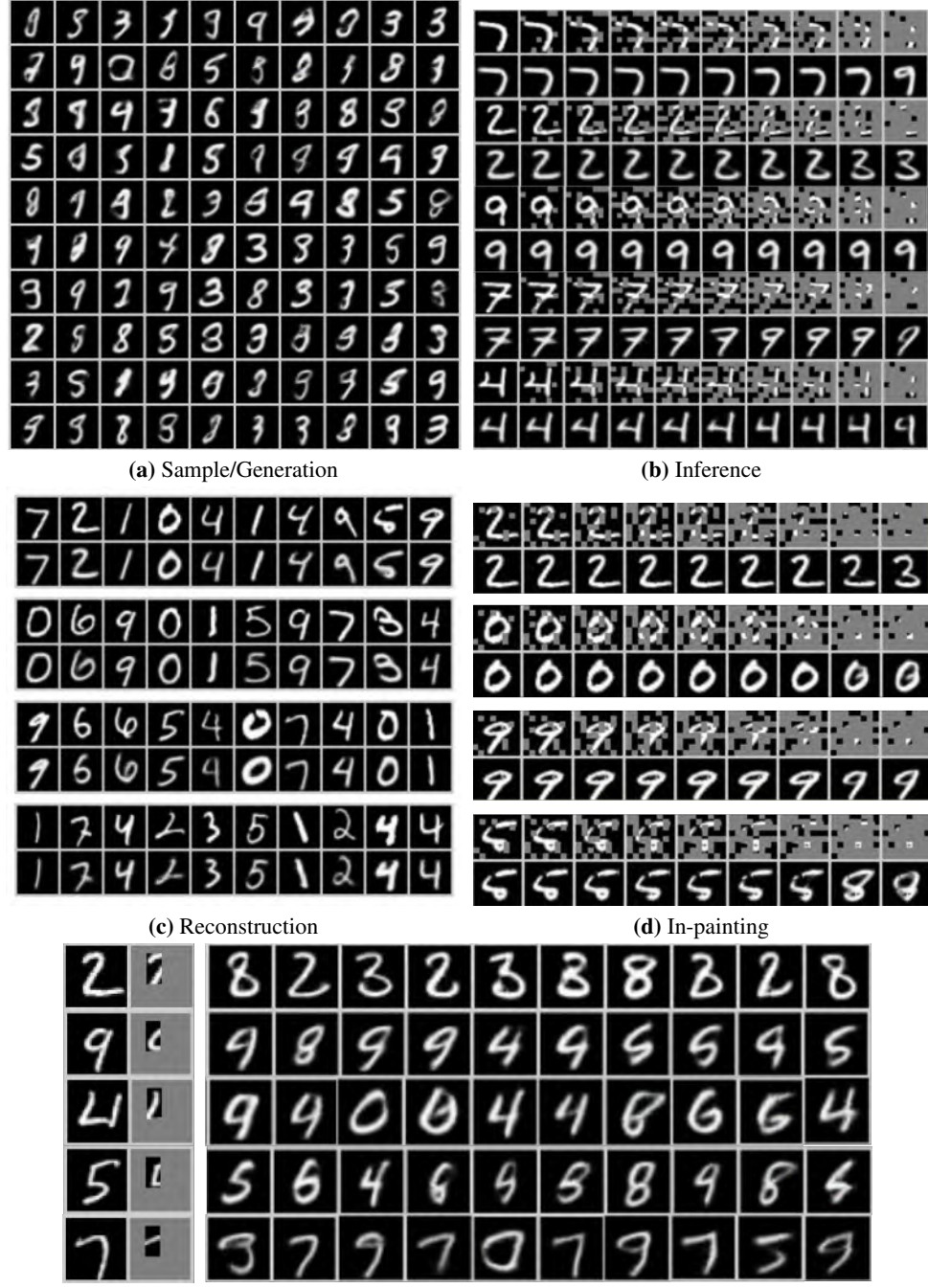

(a) Sample/Generation

(b) Inference

(c) Reconstruction

(d) In-painting

(e) Conditional Sample/Generation

Figure 14: The various model capabilities of BigLearnVAE, illustrated using the MNIST dataset as an example.

