# OpenReview forum: "Big Learning Variational Auto-Encoders"
_ICLR.cc/2024/Conference — Submitted to ICLR 2024_

### Official Review · Reviewer_MAEj · 2023-10-16

**Soundness:** 2 fair
**Presentation:** 2 fair
**Contribution:** 2 fair
**Rating:** 3
**Confidence:** 4

**Summary:**

In this paper, the authors propose big learning VAE to model the inputs from a more general and universal loss perspective. The authors introduce some high-level ideas on big learning, whose objectives should comprise of joint, conditional, and marginal matching tasks. The paper mostly explains the motivation and insights of such big learning concept. The contents also mostly summarizes all different VAE objectives. The final proposed biglearn-VAE is validated on MNIST, FashionMNIST and CelebA, on some inference and in-painting tasks.

**Strengths:**

1. The paper provides and describes many insightful thoughts regarding the universal learning objective for VAE.
2. The core notations and concepts are explained in a self-contained manner.
3. The experiments showcase some capabilities of biglearn-VAE.

**Weaknesses:**

1. Many claims are made on top of the ideal assumptions, while such assumptions are usually hard to achieve. Actually, VAE framework itself is developed based on the fact that $p(x)$ is intractable.
2. Even though you are proposing some universal framework, it's still better to introduce some running examples to facilitate the understanding of your claims. For example, you can use a MNIST example for illustration of different modeling scenarios.
3. During my reading, I do have a feeling that you are more or less re-introducing big learning from a high level, without more concrete or statistical analysis.
4. Section 3.2.3 essentially proposes a more general form of VAE ELBO. However, regarding the specific cases, one still have to reduce the form to more specific ones that we are more familiar with. Moreover, when you claim sth big, readers expect some more convincing and comprehensive experimental results.
5. MNIST, FashionMNIST are well-studied datasets. I'm not sure if they really need large foundation models.
6. Also, the major results you are showing are mostly qualitative, lacking more concrete quantitative ones.

**Questions:**

1.Have you tried your method on some larger-scale datasets?
2. You mentioned text tasks multiple times in the paper, however, there is no text applications in the experimental section?
3. As you mentioned the utilization of large foundation models, maybe it's more convincing if you can give a use case?

---

> ### Author Response · Authors · 2023-11-23
> **Thanks for your comment !**
>
> - **Q1:** Many claims are made on top of the ideal assumptions, while such assumptions are usually hard to achieve.
>
>   **A1:** On one hand, it is an undeniable fact that ideal assumptions are challenging to fully realize in practical applications. However, the value of these ideal assumptions should not be overlooked. They are underpinned by solid theoretical support, providing a necessary theoretical framework that aids in a deeper understanding of the core issues and fundamental laws of this field. By setting idealized conditions, we can more clearly reveal the basic mechanisms behind complex phenomena, laying a solid theoretical foundation for future empirical research and practical application. On the other hand, our method is inspired by insights drawn from foundation models. These foundation models have already demonstrated groundbreaking success and impressive capabilities. Our approach aims to further transfer the model proficiency to the VAE domain, thereby enhancing the performance of VAE models.
>
> - **Q2:** It better to introduce some running examples to facilitate the understanding of your claims.
>
>   **A2:** We have added examples of model capabilities to the revised paper, please refer to Appendix F for details. Using MNIST as an example, we demonstrate the BigLearnVAE's capabilities in ample/generation, inference, reconstruction, in-painting, and conditional sample/generation.
>
> - **Q3:** more or less re-introducing big learning from a high level, without more concrete or statistical analysis.
>
>   **A3:** Reintroducing big learning is intended to facilitate a better understanding of its fundamental concepts, especially for readers who may not have an in-depth familiarity with them. To strengthen our manuscript, we plan to incorporate more detailed statistical evidence and specific examples, which will solidify the theoretical framework we have outlined.
>
> - **Q4:**  Section 3.2.3 essentially proposes a more general form of VAE ELBO. ... reduce the form to more specific ones ....
>
>   **A4:** As defined in Section 3.2.3:
>
>   ​		Where $\mathbb{T} \neq \emptyset$, the analysis under different $(\mathbb{S},\mathbb{T})$ settings is as follows：
>
>   ​			1. When $\mathbb{S} = \emptyset, \mathbb{T} = \mathbb{U}$, BigLearnELBO degenerates to JointELBO.
>
>   ​			2. When $\mathbb{S}=\emptyset,\mathbb{T}\neq\mathbb{U}$ ，BigLearnELBO degenerates to MarginELBO.
>
>   ​         	3. When $\mathbb{S}\neq\emptyset$ ，BigLearnELBO degenerates to ConditioELBO.
>
> - **Q5:** MNIST, FashionMNIST are well-studied datasets. I'm not sure if they really need large foundation models.
>
>   **A5:** Our focus is not on applying foundation models to datasets like MNIST and FashionMNIST, but rather on utilizing the principles of big learning to upgrade traditional VAEs, maintaining the same amount of parameters as traditional VAEs. Moreover, traditional VAEs lack the versatile capabilities exhibited by big learning VAE.
>
> - **Q6：** About the larger-scale datasets, text tasks  and a use case  utilization of large foundation models .
>
>   **A6:** Thank you for suggesting experiments on a larger scale or with text datasets. We plan to explore these in future work. The primary aim of our paper is to enhance VAE through insights drawn from foundation models. Our focus has been on demonstrating this effectiveness using well-known and easily understandable VAE datasets.
>   Given that VAEs haven't been used to train a foundation model, we lack a baseline for comparison. Proposing BigLearnVAE as a novel foundation model, with its demonstrated capabilities, would itself be substantial enough for a separate paper. We consider this an avenue for future

---

### Official Review · Reviewer_DcvD · 2023-10-23

**Soundness:** 2 fair
**Presentation:** 2 fair
**Contribution:** 2 fair
**Rating:** 5
**Confidence:** 2

**Summary:**

This paper studies learning the Variational Auto-encoder (VAE) with the "Big-learning" scheme, which is inspired by the foundation models. Such a learning scheme aims to exploit the large-scale training data with diverse domains exhaustively. The experiments demonstrate the inference capability of such learned VAE.

**Strengths:**

1. The motivation for learning a robust VAE is clear and can contribute to such active research fields for generative models.
2. The organization of the paper is generally well-presented.
3. Although I have only limited experience with such a "big-learning" scheme, I feel like the potential of VAE should be further explored, as many prior works did, and this paper addresses this with a promising direction in a big picture.

**Weaknesses:**

1. I don't have much experience in this particular field, so it is quite difficult for me to follow the paper. The authors intend to apply a robust and general learning scheme for the VAE, with exhaustive data utilization in a multi-modal learning manner (e.g., text and image domains), but some notations seem to be confusing, such as the "joint matching" for marginal distribution p(x).
2. With such a powerful learning scheme, the experiments only demonstrate the inference capability. So I wonder how such a learned VAE performs for generation quality, cross-domain sampling, adversarial robustness, and other standard benchmarks.

**Questions:**

Please see the Weaknesses.

---

> ### Author Response · Authors · 2023-11-23
> **Thanks for your comment !**
>
> Thank you for considering our work to have exploratory potential. Indeed, our method is part of a promising direction within the bigger picture. It is orthogonal to other methods of enhancing VAE, representing  a fresh  and innovative direction.
>
> - **Q1:** ''joint matching'' for marginal distribution p(x).
>
>   **A1:** Generally, in traditional VAEs, ''Joint'' refers to the joint distribution $p(x, z)$ of the data x and the latent variable z, whereas ''Margin'' refers to $ p(x)$. However, in the context of big learning, we approach from the perspective of data where ''Joint'' specifically denotes the distribution of the complete data p(x), and ''Margin'' refers to $p(x_{\mathbb{T}})$. Joint matching refers to $p_{\boldsymbol{\theta}}(\boldsymbol{x}) \longrightarrow q(\boldsymbol{x})$, where $x \sim q(x)$ denotes a complete data sample, that is, $\mathbb{S} \cup \mathbb{T} = \mathbb{L}$. For a more detailed description, you can refer to section 3.2.1 of the manuscript.
>
> - **Q2:** How such a learned VAE performs for generation quality, cross-domain sampling, adversarial robustness, and other standard benchmarks ?
>
>   **A2:** In our manuscript, we demonstrate the model's inference capability, as the inpainting task effectively highlights its ability for conditioned generation. As demonstrated in Appendix C, Figure 8 of the manuscript, our approach showcases a diverse capability in conditional generation. This is particularly notable even when the vast majority of the image area is masked, our model can still generate varied results in terms of hairstyles, facial features, and skin tones, thereby achieving good generative performance. Regarding the cross-domain sampling and related topics you mentioned, we plan to explore these in future work.

---

### Official Review · Reviewer_M8cU · 2023-11-03

**Soundness:** 3 good
**Presentation:** 2 fair
**Contribution:** 2 fair
**Rating:** 5
**Confidence:** 3

**Summary:**

The paper proposed the BigLearn-VAE, inspired by the big learning theorem.  The paper presents extensive analysis and derivation associated with how BigLearn-VAE is motivated, and presents empirical evidences to justify the effectiveness of the method.

**Strengths:**

S1: The paper introduces the BigLearn-VAE, drawing inspiration from the big learning theorem.

S2: It provides a comprehensive analysis and derivation of the underlying principles driving the BigLearn-VAE approach, supported by empirical evidence that substantiates its efficacy.

S3: Empirical evidence shows that the proposed method achieves better ELBO than the vanilla VAE model. The proposed method also can lead to better (visually) generative samples.

**Weaknesses:**

However, the paper seems to lack clarity, which makes the algorithm flow hard to interpret. For example, some of my confusions (owing to the clarity issue) are:

W1: Take for instance, what exactly is the conditional distribution $q(x_t, Z| x_s ) is$, and how to sample from it? I am confused why samples $x_t$ should be dependent on other samples such as $x_s$ rather than independent with each other.

W2: This model seems to be exactly the same as in the conditional VAE, e.g., CVAE in [A]. Can you please distinguish your Biglearn with the work in CVAE [A] ?

W3: It seems the algorithm is only compared with vanilla VAE, whereas there have been many other SOTA version of VAE. The empirical evidence does not address the comparisons with these methods.

W4: An algorithmic flow will probabaly help in terms of clarity when presenting the sampling procedure.

Reference:
[A] Kihyuk Sohn et al., Learning Structured Output Representation using Deep Conditional Generative Models. nips 2015

**Questions:**

Please see the 4 weakness above for my questions. Please correct me during rebuttal if I misunderstood anything.

**Details Of Ethics Concerns:**

None.

---

> ### Author Response · Authors · 2023-11-23
> **Thanks for your comment !**
>
> - **Q1:***  what exactly is the conditional distribution $q\left(x_t, Z \mid x_s\right)$, and how to sample from it? why samples should be dependent on other samples such as rather than independent with each other.
>
>   **A1:** I assume you are referring to the concept of $p( x_{\mathbb{T}} | z,x_{\mathbb{S}} )$, and please correct me if my understanding is incorrect. We begin by sampling the S-ratio and T-ratio from a beta distribution, followed by a random selection of image patches to act as $x_{\mathbb{S}}$ and $x_{\mathbb{T}}$. In the context of Conditional Matching  $p_{\theta}(x_{T} \mid x_{S})$—> $q(x_{T} \mid x_{S})$, the encoder's posterior distribution $p_{\theta}(z \mid x_{S \cup T})$ takes the selected $x_{\mathbb{S}}$ and $x_{\mathbb{T}}$ as inputs to derive the latent variable z. It's important to note that $\mathbb{S} \cup \mathbb{T}$ can differ from $\mathbb{L}$. During the decoding process, the input is conditioned on z and $x_{\mathbb{S}}$, leading to the generation of the predicted $x_{\mathbb{T}}$. Since $x_{\mathbb{S}}$ and $x_{\mathbb{T}}$ originate from the same sample data, they exhibit inherent correlation. For instance, in the CelebA dataset, $x_{\mathbb{S}}$ might represent the left side of a person's face while $x_{\mathbb{T}}$ represents the right side.
>
>
> - **Q2:** Difference between Biglearn and the work in CVAE.
>
>   **A2:** We have thoroughly examined the workings of the CVAE. During its training process, CVAE not only trains $p(y|x)$ but also $p(y|x_{s})$, where x represents the original image, and y denotes the corresponding image segmentation, thereby aligning with the inference task. On one hand, Big Learning VAE targets single-modal tasks and has not yet extended to multi-modal scenarios. On the other hand, the multi-modal setting in big learning is more general, with the training approach of CVAE being a specific case within this broader context.
>
>   For example, with the multi-modal setup, where a data sample $X=(y,x)$ contains both feature $\boldsymbol{x} \in \mathbb{R}^{L \times D}$ and another data modality $\boldsymbol{y} \in \mathbb{R}^{L^y \times D^y}$ with the X-length index set $\mathbb{L}^{\prime}=\left[\mathbb{L}^y, \mathbb{L}\right]$, its any two nonoverlapping source/target index subsets $\mathbb{S}^{\prime}=\left[\mathbb{S}^{\boldsymbol{y}}, \mathbb{S}\right]$ and $\mathbb{T}^{\prime}=\left[\mathbb{T}^y, \mathbb{T}\right] \text { with } \mathbb{S}^{\prime} \subset \mathbb{L}^{\prime}, \mathbb{T}^{\prime} \subseteq \mathbb{L}^{\prime} \text {, and } \mathbb{T}^{\prime} \neq \emptyset$. In $\mathbb{T}^{\prime}$, where $\mathbb{T}^y$ is the universal set and $\mathbb{T}$ is the empty set, BigLearning degenerates into the $p(y|x)$ of CVAE when in $\mathbb{S}^{\prime}$, $\mathbb{S}^y$ is the empty set and $\mathbb{S}$ is the universal set. When in $\mathbb{S}^{\prime}$, $\mathbb{S}^y$ is the empty set and $\mathbb{S}$ is a partial subset, BigLearning degenerates into the $p(y|x_{s})$ of CVAE.
>
> - **Q3:** The empirical evidence does not address the comparisons with many other SOTA version of VAE.
>
>   **A3:** The enhancement of VAE through Big Learning is orthogonal to other methods of improving VAE, which is why we have not made comparisons with other approaches. Our approach is an advancement in the conceptualization of training methodologies, characterized by its higher degree of flexibility. This adaptability allows it to be employed in creating 'big learning' variants of other Variational Autoencoder (VAE) models, exemplified by the development of BigLearn-InfoVAE.
>
> - **Q4:** An algorithmic flow will probabaly help in terms of clarity when presenting the sampling procedure.
>
>   **A4:** The algorithmic flow has been added to the revised paper. Please refer to Appendix D.

---

### Official Review · Reviewer_XfDU · 2023-11-07

**Soundness:** 3 good
**Presentation:** 2 fair
**Contribution:** 2 fair
**Rating:** 5
**Confidence:** 3

**Summary:**

The authors present "Big Learning" variational autoencoders that attempt to train VAEs and their respective marginals and conditional distributions simultaneously.  The aim is to demonstrate that such an approach is better able to handle incomplete data than vanilla VAEs.  Experiments are designed to validate these claims.

**Strengths:**

The key idea is simple, and it extends the capabilities of VAEs -- effectively transferring computational time from inference to training time.

**Weaknesses:**

The novelty of this work is a bit limited over top of Cong and Zhao, 2022.

The main idea is relatively simple, but the explanation of that idea is a bit rambling.  In particular, I found Definition 1 initially confusing. The presentation would benefit from a clear problem statement.

The performance comparison in Figure 2 shows comparable performance to the vanilla VAE (which honestly isn't that surprising given the training setup). These are really the only quantitative results.  To me, it makes more sense to compare against the sampling methods that provide similar capabilities to the big learned model.  In a sense, this approach is pushing the computational load to the training phase instead of the inference phase, which makes sense, but you'd still want to verify that the resulting model has competitive performance on the proposed task against existing approaches.

Overall, I found the experiments a bit underwhelming and the details of the experimental setup/evaluation are scant.

**Questions:**

See the above.

Additional questions:
-  What is the difference in training time between the vanilla VAE and this approach?


Minor typos/suggestions:
-  The citation style is not correct.  Please use parenthetical citations instead of in-text citations to make the paper more readable.
-  "-in-paining" -> "in-painting"
-  "review the preliminary Variational"
-  "be selected base on"
-  "space is important and many works" -> "space is important, and many works"
- "can not" -> "cannot"
- "one need two foundation"

---

> ### Author Response · Authors · 2023-11-23
> **Thanks for your comment !**
>
> - **Q1:** About the novelty.
>
>   **A1:** Our method is based on big learning, a concept initially proposed for foundation models. We have transferred its demonstrated flexibility and practicality to the traditional VAE framework, embodying a pattern of knowledge feedback.
>
> - **Q2:** What is the difference in training time between the vanilla VAE and this approach?
>
>   **A2:** The training time required for vanilla VAE and BigLearnVAE, as well as the curves showing the change in loss during the training process, have been added to the revised paper. Please refer to Appendix E. Under the same model size and experimental settings, the training loss and training time changes for vanilla VAE and BigLearnVAE are shown in Appendix E Figure 13. Theoretically, with the same network architecture, the training of one generation in a traditional VAE and our method are essentially the same, with comparable computational costs. Our method only incurs additional time for sampling. As can be seen from Figure 1, the convergence speed and total training time of BigLearnVAE are roughly the same as those of the vanilla VAE.

---

### Official Review · Reviewer_b3xg · 2023-11-10

**Soundness:** 1 poor
**Presentation:** 2 fair
**Contribution:** 2 fair
**Rating:** 3
**Confidence:** 5

**Summary:**

The paper presents a modification to the Variational Autoencoder (VAE) framework by incorporating conditional encoding and decoding processes capable of handling incomplete data inputs. Here's a summary of the modifications:

Conditional Encoder: Instead of requiring a complete data point 'x', the new encoder can work with partial data. It is designed as a transformer architecture, where missing values in 'x' are masked. A special 'CLS' token is used in the final position to generate a probability distribution over the latent variable 'z'.
Conditional Decoder: The decoder is adapted to accept both the latent variable 'z' and a partial 'x'. It then reconstructs the missing components of 'x' sequentially. This is achieved through a transformer setup with masking for the missing values, using an index-based approach to handle the decoding of missing elements of 'x'.
Additionally, the authors introduce a distribution over which positions in the input are missing. The decoder is trained across all possible combinations of missing and present data.

**Strengths:**

It is an interesting and novel idea to train the VAE across all possible combinations of missing and present data, thereby learning a comprehensive generative process even with incomplete inputs. This approach enhances the model's capability to handle and predict missing data within a given dataset.

**Weaknesses:**

The primary contribution of the paper is section 3.2.2 which is filled with inconsistencies
 1) > $\text{The marginal } p_\theta(x_T, z) \text{ is readily derived from the joint } p_\theta(x, z) \text{ via index selection with } T; \text{ with specific } T = L, \text{ the MarginELBO in (4) reduces to the JointELBO in (1);}$

Note that none of these group of distributions are consistent wrt each other. Consider a 2 dimensional x = (x1, x2), then
$$p(x1| mask, x2, z)p(x2|mask, mask, z) \neq p(x2|x1, mask, z)p(x1|mask, mask, z)$$

So, unlike expected by the authors, this approach doesn't allow for modelling arbitrary conditional distributions $p(x_T|x_S, z)$

 2) > $\text{Both optima have already been modeled in the parameterized } p_\Phi(z | x_{S'}).$

This is incorrect. The definition of optimal $q_M(z|x_T)$, that is, $p_\theta(z|x_T)$, is a distribution that follows from Bayes' rule, that is
$$p_\theta(z|x_T) \propto p_\theta(x_T|z)p(z) $$
$p_\Phi(z | x_{T})$ doesn't model this distribution

3) The notations used throughout the paper are very confusing. The same q had been used to represent the emprical data distribution, the posteriors and the distribution over indices q(S,T).

4) The writing of the paper can be improved. Primarily, it is not a good idea to present the paper as a special case of big-learning [1] (an unknown unpublished/rejected work that claims to be all-encompassing). I tried reading the big-learning paper but there were too many errors in that paper to go-through.

[1] Yulai Cong and Miaoyun Zhao. 2023. "Big Learning: A Universal Machine Learning Paradigm?" [Online]. Available at: https://openreview.net/forum?id=UfFXUfAsnPH.

**Questions:**

1) How is q(S,T) chosen in equation (7)?
2) What is index-based decoding mentioned at the beginning of page 6?

---

> ### Author Response · Authors · 2023-11-23
> **Thanks for your comment !**
>
> - **Q1:** Consider a 2 dimensional $x = (x1, x2)$, then
>   $$
>   p(x1|\text{mask}, x2, z)p(x2|\text{mask}, \text{mask}, z) \neq p(x2|x1, \text{mask}, z)p(x1|\text{mask}, \text{mask}, z).
>   $$
>   **A1:** Joint learning cannot automatically provide marginal/conditional matching, for this reason we explicitly introduce big learning. By modeling itself, it won't lead to consistency, that's why we introduce big learning to bring about consistency through learning. For example, under the likelihood architecture, $log q(x1, x2) = log q(x1|x2)q(x2) = log q(x2|x1)q(x1)$, where $q$ represents the data distribution. Big learning explicitly and simultaneously matches the joint distribution under both $log q(x1|x2)q(x2)$ and $log q(x2|x1)q(x1)$ distributions. Ideally, when the above two distributions match well, we can obtain $p(x1|\text{mask}, x2, z)p(x2|\text{mask}, \text{mask}, z)=p(x2|x1, \text{mask}, z)p(x1|\text{mask}, \text{mask}, z)$. In practice, since the optimization goal of both of the above distributions is to match the joint distribution, it is expected that the results will not differ significantly.
>
> - **Q2:** ...$q_M\left(z \mid x_T\right)$, that is $p_\theta\left(z \mid x_T\right)$, ... follows from Bayes' rule $p_\theta\left(z \mid x_T\right) \propto p_\theta\left(x_T \mid z\right) p(z)$, $p_{\Phi}\left(z \mid x_T\right)$ doesn't model this distribution.
>
>   **A2:** In the $\Phi$ space, $p_{\Phi}\left(z \mid x_T\right)$ is our parameterization method, and we use the $p_{\Phi}\left(z \mid x_T\right)$ to model the data distribution $q_M\left(z \mid x_T\right)$.
>
> - **Q3:** The same q had been used to represent the emprical data distribution, the posteriors and the distribution over indices q(S,T).
>
>   **A3:** We follow the notation in the VAE paper to represent the empirical data distribution and the posteriors distribution, with ELBO being identical to $ KL[q(x)q(z|x) \mid \mid p(x, z)] $. Generally, in VAEs, the generative process is denoted by $p$,  while the data and posterior are represented by $q$. Therefore, here we choose $q$ to represent the $(S,T)$ distribution.
>
> - **Q4:** ...it is not a good idea to present the paper as a special case of big-learning.
>
>    **A4:** We will carefully revise the manuscript to enhance its clarity. Our method is essentially inspired by the big learning method. However, our focus is on the VAE domain. We verify that the big learning idea works effectively.
>
> - **Q5:** How is $q(S,T)$ chosen in equation (7).
>
>   **A5:** We sample the S-ratio and T-ratio from a beta distribution, specifically choosing parameters $(\beta_1, \beta_2)=(0.5,3)$ for the S-ratio and $(\beta_1, \beta_2)=(3,0.5)$ for the T-ratio. This parameterization for the S-ratio ensures that it is more likely to take smaller values, while the chosen parameters for the T-ratio make it more likely to assume larger values. Subsequently, we perform a random selection of image patches to serve as $x_{\mathbb{S}}$ and $x_{\mathbb{T}}$, where notably, $\mathbb{S} \subset \mathbb{L}$, $\mathbb{T} \subseteq \mathbb{L}$, $\mathbb{T} \neq \emptyset$. Through this diverse combination of S-ratio and T-ratio, the model is trained to integrate both generative and inference capabilities.
>
> - **Q6:**  What is index-based decoding mentioned at the beginning of page 6 ?
>
>   **A6:** As illustrated in Fig. 1b of the manuscript, we input into the decoder both the encoded variable z and the selected  $x_{\mathbb{S}}$, chosen using the method described in question 1. The decoder then outputs the corresponding selected image patches $x_{\mathbb{T}}$.

---

### Meta-Review · Area_Chair_fjJA · 2023-12-09

**Metareview:**

The paper proposes to extend the variational autoencoder (VAE) framework to accomodate for transformers models in the encoder and decoder as to deal with possibly missing inputs and outputs. The resulting architecture (BigLearnVAE) is used in a number of representation learning tasks on image data.

In their reviews and during the rebuttal, reviewers highlighted several concerns about the presentation of the work and its content. Specifically, they questioned several claims about the heuristic nature of the imputation/marginalization task, about the consistency across marginals, and novelty of the architecture of BigLearnVAE, w.r.t. previous VAE variants that were applying conditioning over inputs and outputs to deal with missing values.

In the rebuttal, authors updated the paper with some examples, and pointed to the "big learning framework" as a reason for novelty. From the current version of the paper it is however hard to understand how this big learning framework differs from a normal transformer architecture and therefore correctly position this submission in the large VAE literature. The other reviewers' questions concerning consistency remain overall unanswered during the rebuttal.

The paper is therefore rejected.

**Justification For Why Not Higher Score:**

All reviewers agreed that the contribution is not precisely presented and rigorously compared and contrasted with other VAE variants.

**Justification For Why Not Lower Score:**

N/A

---

### Decision · Program_Chairs · 2024-01-16

Reject